# De novo production of protoberberine and benzophenanthridine alkaloids through metabolic engineering of yeast

Xiang Jiao [1], Xiaozhi Fu [1], Qishuang Li [2], Junling Bu[2], Xiuyu Liu[2], Otto Savolainen[1,3], Luqi Huang[2] ✉, Juan Guo [2] ✉, Jens Nielsen [1,4] ✉ & Yun Chen [1] ✉

Protoberberine alkaloids and benzophenanthridine alkaloids (BZDAs) are subgroups of benzylisoquinoline alkaloids (BIAs), which represent a diverse class of plant-specialized natural metabolites with many pharmacological properties. Microbial biosynthesis has been allowed for accessibility and scalable production of high-value BIAs. Here, we engineer *Saccharomyces cerevisiae* to de novo produce a series of protoberberines and BZDAs, including palmatine, berberine, chelerythrine, sanguinarine and chelirubine. An ER compartmentalization strategy is developed to improve vacuole protein berberine bridge enzyme (BBE) activity, resulting in >200% increase on the production of the key intermediate (*S*)-scoulerine. Another promiscuous vacuole protein dihydrobenzophenanthridine oxidase (DBOX) has been identified to catalyze two-electron oxidation on various tetra-hydroprotoberberines at N7-C8 position and dihydrobenzophenanthridine alkaloids. Furthermore, cytosolically expressed DBOX can alleviate the limitation on BBE. This study highlights the potential of microbial cell factories for the biosynthesis of a diverse group of BIAs through engineering of heterologous plant enzymes.

The misuse and abuse of antibiotics in human and livestock settings has led to the emergence of antimicrobial resistance, which poses a serious threat on global public health, as highlighted by the World Health Organization in 2015[1]. As an alternative approach, the exploration of phytogenic natural products with antimicrobial properties has gained much attention. Protoberberine alkaloids and benzophenanthridine alkaloids (BZDAs) are two classes of such natural products that have attracted increasing interest due to their bioactivity and safety against microorganisms and viruses[2]. Both protoberberines and BZDAs belong to the big family of benzylisoquinoline alkaloids (BIAs), which encompasses more than 2500 structurally diverse compounds known for their pharmacological properties[3]. Within this family, berberine stands out as a prominent protoberberine alkaloid, demonstrating efficacy against numerous pathogenic bacteria and viruses[4]. Furthermore, many clinical trials on berberine have been conducting to evaluate its pharmacological properties, including its anti-hyperlipidemic[5], anti-diabetic[5], anti-inflammation[6], and the potential to prevent atherosclerosis[7]. Another notable protoberberine alkaloid, palmatine exerts similar antioxidant and anti-inflammatory properties with berberine and has been proposed as a promising DNA phototherapy drug[8]. Regarding BZDAs, one notable BZDA-containing product, Sangrovit®, mainly derived from bloodroot

[1]Department of Life Sciences, Chalmers University of Technology, Kemivägen 10, SE-412 96 Gothenburg, Sweden. [2]State Key Laboratory for Quality Ensurance and Sustainable Use of Dao-di Herbs, National Resource Center for Chinese Materia Medica, China Academy of Chinese Medical Sciences, 16 Neinanxiaojie, Dongcheng district, Beijing, China. [3]Chalmers Mass Spectrometry Infrastructure, Chalmers University of Technology, Kemivägen 10, SE-412 96 Gothenburg, Sweden. [4]BioInnovation Institute, DK-2200 Copenhagen N, Denmark. ✉e-mail: huangluqi01@126.com; guojuanzy@163.com; nielsenj@chalmers.se; yunc@chalmers.se

plants *Macleaya cordata* and *Sanguinaria canadensis*, has been developed as a safe natural feed additive[9], exhibiting benefits in terms of improving growth performance and health in livestock farming and aquaculture[10,11]. Beyond their antimicrobial and animal growth-promoting properties, many BZDAs have demonstrated great potential for medical importance. For instance, sanguinarine has been identified as a potential anticancer drug due to its ability to stabilize human telomeric G-quadruplex DNA[12,13]. Chelerythrine, a well-known protein kinase C inhibitor, and macarpine, which shows an anti-proliferative effect on several cancer cell lines, hold promise for various applications[14,15]. Notably, chelerythrine has been recently reported as a multi-purpose adjuvant for the treatment of COVID-19[16,17]. Although isolation of these compounds from plants is possible, it is challenging to meet the increasing demand via large-scale plant cultivation and therefore alternative methods, i.e., microbial biosynthesis for reliable and scalable supply of these products are urgently desired. The key to microbial-based production is to improve the performance of enzymes thus ensuring an efficient biocatalytic process.

In plants, the committed step for protoberberines and BZDAs biosynthesis is stereoselective conversion of (S)-reticuline ((S)-RET) to (S)-scolerine ((S)-SCO), catalyzed by berberine bridge enzyme (BBE) (Fig. 1). This flavoprotein oxidase has been identified as a key rate-limiting enzyme in the biosynthesis of protoberberines and BZDAs. However, enhancing the catalytic efficiency of BBE in heterologous microbial hosts is challenging, likely due to the differences in micro-environments between plants and microbes. Previous attempts to

increase BBE expression level via plasmid or multiple chromosomal integration did not significantly improve its conversion efficiency in microbial hosts[18–20]. In vivo assays demonstrated that BBE is expressed via the secretory pathway and localized to the plant vacuole[21,22], implying the importance of post-translational modifications and trafficking processes for its activity. Since the N-terminal signal sequence is usually essential to direct the protein to undergo such processing steps, the common strategies to improve the folding or stability of plant enzymes through N-terminus engineering would not be feasible for BBE optimization. Indeed, the N-terminal truncations of BBE resulted into its decreased activity in yeast[18,23]. Therefore, improving BBE efficiency in heterologous microbial hosts such as *S. cerevisiae* remains challenging.

The final step for protoberberines and BZDAs biosynthesis involves the conversion of tetrahydroprotoberberines and dihydrophenanthridines by additional flavoproteins, including (S)-tetrahydroprotoberine oxidase (STOX) or dihydrobenzophenanthridine oxidase (DBOX). Although there were reports on the production of tetrahydroprotoberberines such as (S)-tetrahydropalmatine[24] and dihydrophenanthridines such as dihydrosanguinarine[18,19], microbial de novo production of protoberberines and BZDAs such as palmatine, chelerythrine and sanguinarine in yeast have been absent. One major challenge is the functional expression of the final catalytic flavoprotein in yeast. There have been several flavoenzymes reported to catalyze this type of reaction leading to two-electron (one double bond formation) or four-electron oxidations (two double bonds formation) in

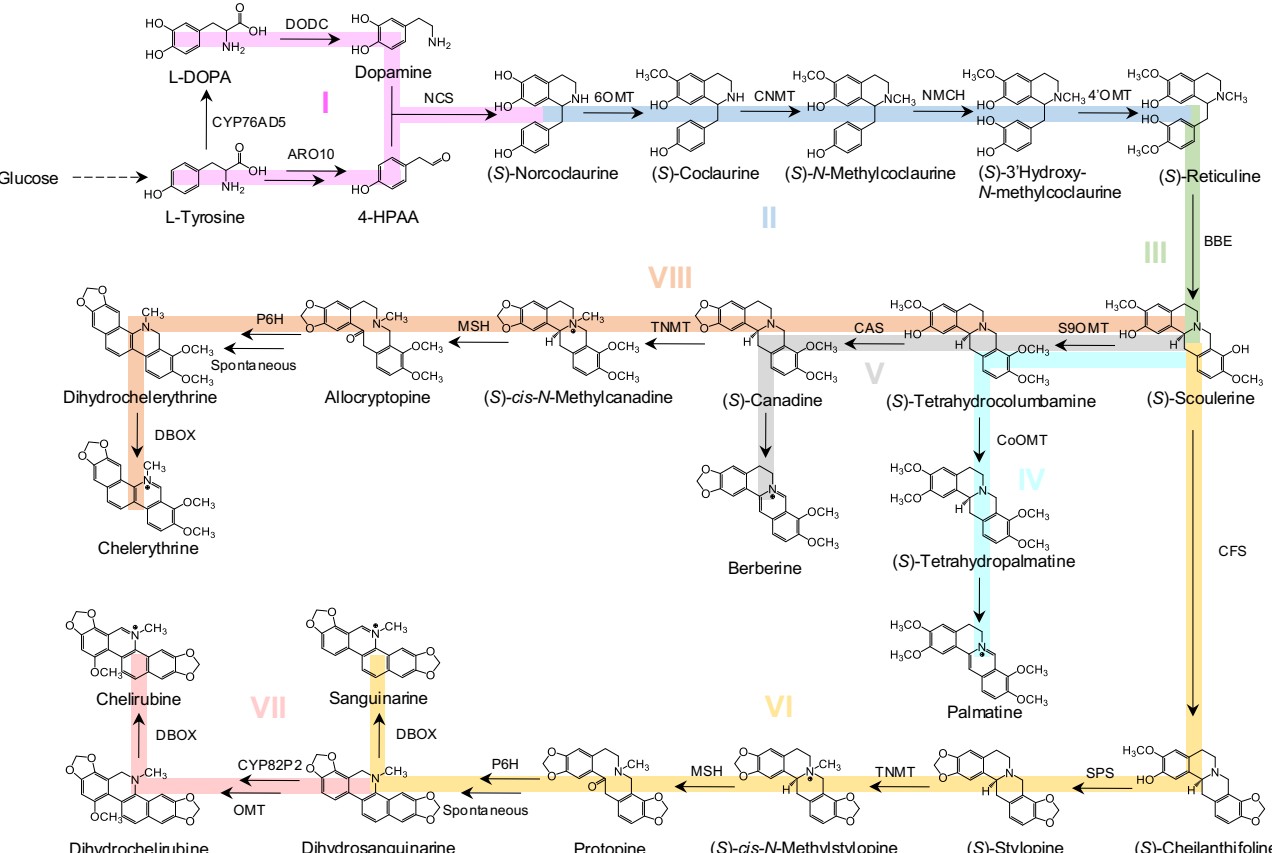

**Fig. 1 | Schematic presentation of reconstructing the biosynthetic pathway for de novo production of protoberberines and BZDAs in yeast.** To enable the reconstruction of the complete pathway more accessible, it is divided into eight modules as different colors highlighted. Module I (pink) initiates from the condensation of dopamine and 4-HPAA to generate the first BIA compound (S)-norcoclaurine ((S)-NOR), which was sequentially converted into the core intermediates

(S)-reticuline ((S)-RET) and (S)-scoulerine ((S)-SCO) in module II (blue) and module III (green). Module IV (sky blue), module V (gray), module VI (yellow) and module VIII (orange), diverged from the branch point intermediate (S)-SCO, aim to produce palmatine, berberine, sanguinarine, and chelerythrine, respectively. Module VII (rose) focuses on extending the pathway from sanguinarine to biosynthesize Chelirubine.

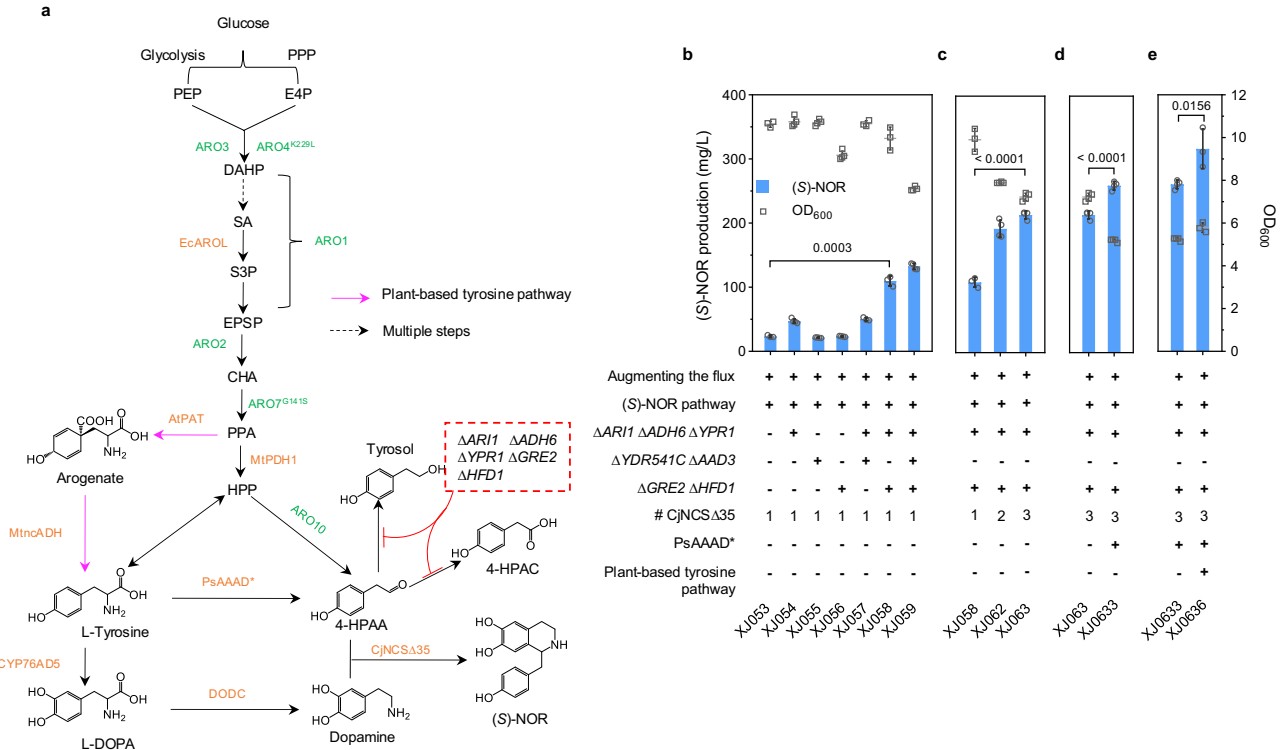

**Fig. 2 | Optimizing (S)-NOR production in yeast. a** Biosynthetic pathway from glucose to generate (S)-NOR in yeast. Color scheme of pathway: green, over-expression of endogenous genes or mutants; orange, introduction of heterologous genes; purple arrows, plant-based tyrosine pathway; black dashed arrow, multiple steps; red, gene deletion. EcAROL, shikimate kinase; MtPDH1, prephenate dehydrogenase; AtPAT, prephenate aminotransferase; MtncADH, noncanonical arogenate dehydrogenase; PsAAAD*, aromatic amino acid decarboxylase mutant. PPP, pentose phosphate pathway; E4P, erythrose 4-phosphate; PEP, phosphoenolpyruvate; DAHP, 3-deoxy-D-arabino-heptulosonate 7-phosphate; SA, shikimic acid; S3P, shikimate 3-phosphate; EPSP, 5-enolpyruvylshikimate-3-phosphate; CHA, chorismate; PPA, prephenate; HPP, 4-hydroxyphenylpyruvate; 4-HPAA, 4-hydroxyphenylacetaldehyde; Tyrosol, 2-(4-Hydroxyphenyl) ethanol; 4-HPAC, 4-Hydroxyphenylacetic acid. (S)-NOR titers and final $OD_{600}$ in engineered strains with **b** the removal of potential aldehyde reductases or dehydrogenases; **c** increasing more copies of CjNCSΔ35; **d** introducing aromatic amino acid synthase and **e** plant-based tyrosine pathway. At least three biologically independent colonies were grown for 72 h in 20 mL minimal media with 20 g/L glucose. Augmenting the flux indicates overexpression of yeast native *ARO1, ARO2, ARO3*, and expression of EcAROL, MtPDH1, ARO4[K229L] and ARO7[G141S]. (S)-NOR pathway indicates the introduction of optimal CYP76AD5, DODC and CjNCSΔ35. Significance was calculated using two-tailed *t*-test. Data are presented as mean ± standard deviations (*n* = 3 or 4 biologically independent samples). Source data are provided as a Source Data file.

vitro, such as AmSTOX[25], CjTHBO[26] and PsFADOX5[27]. However, the in vivo activity of the corresponding enzymes was neither confirmed in plants nor observed in yeast. Recently, the BZDAs biosynthetic pathways were characterized in *Macleaya cordata*[28], which provides new opportunities for microbial production of protoberberines and BZDAs.

In this work, we develop a yeast-based platform for de novo production of a series of protoberberines and BZDAs. We first construct platform strains for producing the key intermediate (S)-SCO, its titer reaching up to 113 mg/L through a series of modifications involving ER re-localization of vacuolar BBE, ER expansion, and the alleviation of $H_2O_2$ toxicity in the ER. By leveraging this (S)-SCO platform yeast, we extend the pathway to produce a wide spectrum of protoberberines and BZDAs, including dehydroscoulerine, columbamine, palmatine, berberine, chelerythrine, sanguinarine and chelirubine. Specifically, we functionally express additional vacuolar oxidase McDBOX2 from *M. cordata*, capable of removing two electrons to form one N7＝C8 band in vivo. The introduction of this enzyme enables the complete reconstruction of heterologous BZDAs pathway in yeast. However, its expression in vacuole causes some post-translational processing competitions with BBE, this issue can be alleviated to some extent by cytosolic expression of McDBOX2. Furthermore, we demonstrate that McDBOX2 could catalyze many tetrahydroprotoberberines to produce corresponding 13,14-dihydroprotoberberines in yeast. Considering flavoenzyme-catalyzed

oxidation as a common step involved in BIAs biosynthesis, the strategies developed in this study provide insights towards generalizing the engineering of *S. cerevisiae* for de novo producing a broader class of medicinal BIAs.

## Results

### Constructing a platform strain for (S)-norcoclaurine production

In this study, we constructed yeast strains for de novo production of protoberberines and BZDAs, such as berberine, palmatine, chelerythrine, sanguinarine and chelirubine. To streamline the reconstruction process, the whole biosynthetic pathway was divided into eight modules as depicted in Fig. 1 (An overview of constructed strains can be found in Supplementary Fig. 1 and Supplementary Data 1).

To achieve efficient (S)-norcoclaurine ((S)-NOR) production, we first focused on selecting an optimal tyrosine hydroxylase for dopamine biosynthesis (Fig. 2a). Several candidate enzymes reported previously to catalyze the hydroxylation of tyrosine were evaluated, including CYP76AD1* and CYP76AD5 from the sugar beet *Beta vulgaris*[29,30], C3H from *Arabidopsis thaliana* and *Zea mays*[31], HpaB and HpaC from *Pseudomonas aeruginosa* and *Salmonella enterica*[32]. Among these, CYP76AD5 led to at least three-fold higher dopamine production than the other candidates (Supplementary Fig. 2a). Next, we compared several candidates to identify a superior norcoclaurine synthase (NCS) enzyme for (S)-NOR biosynthesis: CyNCS12 from *Corydalis yanhusuo*, PsNCS3 from *Papaver somniferum*, PbNCS5 from

*Papaver bracteatum*, StNCS2 and StNCS4 from *Stephania tetrandra*, BtNCS from *Berberis thunbergii*, CjNCS from *Coptis japonica*. N-terminal truncation of CjNCS has been reported to improve its expression level[33], we thus truncated 24, 29 and 35 amino acids on the N-terminus of CjNCS to generate CjNCSΔ24, CjNCSΔ29 and CjNCSΔ35, respectively. All these NCS candidates were compared in a dopamine producing strain XJ001, with CjNCSΔ35 resulted in the highest (S)-NOR titer (Supplementary Fig. 2b).

Subsequently, we constructed a reference strain XJ023 by augmenting the flux towards tyrosine biosynthesis based on our previous study[34], including overexpression of yeast native genes *ARO1*, *ARO2*, *ARO3*, and expression of EcaroL from *E. coli*, MtPDH1 from *Medicago truncatula*, and two mutants Aro4^K229L and Aro7^G141S. By introducing the optimal enzyme CYP76AD5, DODC and CjNCSΔ35 into strain XJ023, the resultant strain XJ053 produced 23.4 mg/L (S)-NOR (Fig. 2b). However, large amounts of dopamine (164.3 mg/L), 4-hydroxyphenylacetic acid (4-HPAC, 34.8 mg/L) and tyrosol (281.7 mg/L) were also detected in XJ053 (Supplementary Fig. 3), indicating that most 4-hydroxyphenylaldehyde (4-HPAA) flux flew towards by-product formation. To address this issue, three different strategies were tested. The first strategy involved encapsulating CjNCSΔ35 and Aro10 (encoding phenylpyruvate decarboxylase) or CjNCSΔ35, Aro10, and DODC within a bacterial nanocompartment derived from *Myxococcus xanthus*[35]. However, both attempts only showed a marginal increase on (S)-NOR production (6% *p* = 0.0117 and 13% *p* = 0.0002) compared with the control (Supplementary Fig. 4a–c). The second strategy employed peroxisome-based compartmentalization. Unanticipatedly, targeting CjNCSΔ35 or its combination with Aro10, DODC, and CYP76AD5 into the peroxisome resulted in a significant decrease on (S)-NOR production in all cases (Supplementary Fig. 5). This contradicted a recent report but could be attributed to the differences on NCS expression levels, precursor production, or strain backgrounds[36]. For the third strategy, we removed some competition from potential aldehyde reductases or dehydrogenases on the substrate 4-HPAA, such as *ARI1*, *ADH6*, *YPR1*, *YDR541C*, *AAD3*, *GRE2* and *HFD1*[37]. Although disruption of all seven genes slightly increased (S)-NOR production in strain XJ059, it also led to a 24% (*p* = 0.0003) decrease on biomass yield compared with strain XJ058 without deletion of *YDR541C* and *AAD3* (Fig. 2b). To evaluate if the expression level of downstream enzyme CjNCSΔ35 was limited, strain XJ058 was selected to introduce more copies of CjNCSΔ35. Two additional copies of CjNCSΔ35 (strain XJ063) prompted (S)-NOR titer to 214.8 mg/L, a 2-fold increase compared with strain XJ058 (Fig. 2c). Further integration of more copies did not increase (S)-NOR titers (Supplementary Fig. 6a). Intriguingly, dopamine, tyrosol and 4-HPAC titers remained at around 100 mg/L between second and five copies integration (Supplementary Fig. 6b), indicating that there might be some uncharacterized aldehyde reductases or dehydrogenases in yeast, which exhibits a relative higher affinity on 4-HPAA compared to CjNCSΔ35.

To further improve the availability of 4-HPAA, we employed aromatic amino acid decarboxylase (AAAD) to channel more flux towards 4-HPAA (Fig. 2a). Apart from endogenous Aro10, three aromatic amino acid (AAS) homologies, including PcAAS from *Petroselinum crispum*[38], RrAAS form *Rhodiola Salidroside*[39], and a mutated PsAAAD* (Y350F) from *P. somniferum*[40], were compared in strain XJ063 expressing three copies of CjNCSΔ35. All four resultant strains produced more (S)-NOR than the control strain XJ063, with strain XJ0633 containing PsAAAD* yielding the highest titer (295.5 mg/L), a 38% (*p* < 0.0001) increase relative with strain XJ063 (Fig. 2d and Supplementary Fig. 7).

Expanding the possibility of rewiring tyrosine biosynthesis, we further explored a plant-based tyrosine pathway in yeast including a PPA aminotransferase (PAT), and an arogenate dehydrogenase (ADH) (Fig. 2a and Supplementary Fig. 8a). To do so, three PATs (AtPAT from *A. thaliana*, BvPAT1 and BvPAT2 from *B. vulgaris*)[41,42] and four ADHs (BvADHα from *B. vulgaris*, MjADHα from *Mirabilis jalapa*, MtncADH

from *M. truncatula*, and GmncADH from *Glycine max*)[43,44] were tested in a *TYR1* deletion strain, which exhibited a tyrosine auxotrophic phenotype. Strikingly, five combinations of PAT and ADH (AtPAT and BvADHα, AtPAT and MtncADH, BvPAT1, and BvADHα, BvPAT1 and MtncADH, BvPAT2, and MtncADH) restored the growth of *TYR1*-deficiency strain, suggesting the function of the plant-based tyrosine pathway in yeast (Supplementary Fig. 8b). These five specific enzyme combinations were then evaluated in strain XJ0633 expressing PsAAAD*. The BvPAT2 and MtncADH combination displayed a negative impact on (S)-NOR titer, whereas the other four led to the improved (S)-NOR titers, especially with AtPAT and MtncADH combination exhibiting a 22% (*p* = 0.0156) increase (XJ0636, 315.9 mg/L), compared to the control strain XJ0633 (Fig. 2e and Supplementary Fig. 8c). Therefore, XJ0636 was utilized as the (S)-NOR-producing platform strain for subsequent experiments.

## Constructing a platform strain for producing (S)-reticuline

Module II focused on modifying (S)-NOR to produce downstream (S)-reticuline ((S)-RET), which sequentially catalyzed by 6-O-methyltransferase (6OMT), coclaurine N-methyltransferase (CNMT), N-methylcoclaurine 3′-hydroxylase (NMCH) and 4′-O-methyltransferase (4′OMT) (Fig. 3a). To identify the optimal enzyme combination of the first three steps, Ps6OMT, PsCNMT from *P. somniferum*, EcNMCH from *Eschscholzia californica*, along with alternative Cy6OMT, CyCNMT and CyNMCH from *C. yanhusuo* were tested in a simple background strain XJ040, harboring a cytochrome P450 reductase and a 4′OMT from *P. somniferum* (PsCPR and Ps4′OMT) by measuring (S)-RET titers. The combination, consisting of Ps6OMT, PsCNMT and CyNMCH, led to the highest (S)-RET production and CyNMCH performed generally better than EcNMCH whereas others did not display significant difference (Supplementary Fig. 9). When reverse-sequentially expressing the second copy of module II enzymes, only CyNMCH overexpression displayed a significant increase on (S)-RET titer (Supplementary Fig. 10), indicating that the hydroxylation catalyzed by NMCH is a rate-limiting step.

In eukaryotes, the most ubiquitous P450s belong to class II of P450 superfamily[45], which are localized to the membrane of ER and composed of two components, a P450 domain and a FAD/ FMN containing NAD(P)H CPR domain, shuttling the electrons to P450. To improve the efficiency of CyNMCH, we subsequentially explored coupling different cytochrome P450 reductase (CPR) to transfer the electrons. Apart from PsCPR, ATR1 and ATR2 from *A. thaliana*, were also compared in the (S)-NOR platform strain XJ0636 with other four optimal enzymes (Ps6OMT, PsCNMT, CyNMCH, and Ps4′OMT) for (S)-RET biosynthesis. Both ATR1 (273.6 mg/L) and ATR2 (255.7 mg/L) led to (S)-RET titer improvement, compared to PsCPR (222.3 mg/L), indicating better pairing of ATR1/2 with CyNMCH (Fig. 3b). However, significant accumulation of intermediates 3′hydroxy-N-methylcoclaurine and (S)-N-methylcoclaurine were observed in the best strain XJ0675 containing ATR1 (Supplementary Fig. 11). We thus increased the copy number of CyNMCH and Ps4′OMT in strain XJ0675 by using a landing pad system for precise multicopy gene integration[46]. Briefly, the landing pads, carrying single nonnative gRNA cutting site, were inserted when knocking out five genes *ARI1* (LP3.T7), *ADH6* (LP3.T7), *YPR1* (LP3.T7), *GRE2* (LP1.T8) and *HFD1* (LP4.T9). The resulting strain XJ0691, harboring 2 copies of Ps4′OMT and 5 copies of CyNMCH, produced 425.2 mg/L (S)-RET, representing an 85% (*p* < 0.0001) increase relative to the control strain XJ0675 after 72 h in shake flask cultivation with 20 g/L glucose (Fig. 3c).

## Fine-tuning BBE to improve (S)-scoulerine titer

In module III, we focused on BBE which catalyzes the oxidative cyclization of the N-methyl moiety of (S)-RET into the berberine bridge of (S)-scoulerine ((S)-SCO), a major branch intermediate leading to the biosynthesis of protoberberines and BZDAs[47] (Fig. 4a). To select an

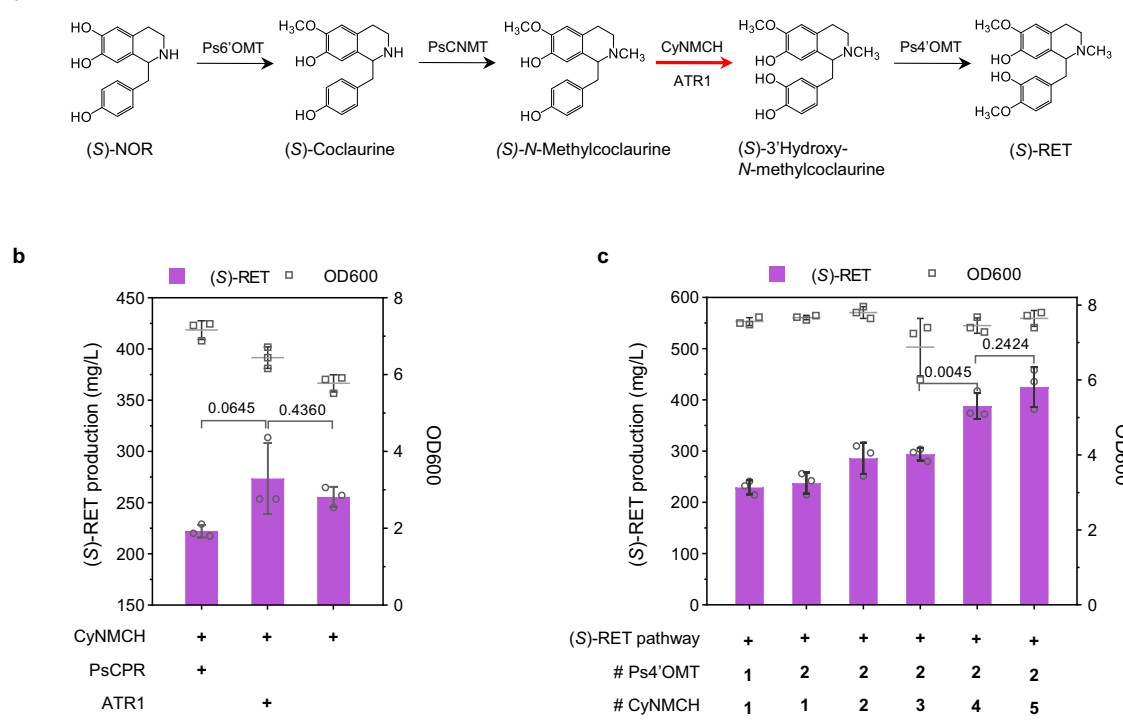

**Fig. 3 | Extending the biosynthetic pathway from (S)-NOR to produce (S)-RET.**
**a** Schematic presentation of module II. P450 NMCH catalyzes the rate-limiting step of hydroxylation of N-methylcoclaurine. CPR could shuttle two electrons to P450 for its catalyzation. Optimal coupling of alternative P450 and CPR enables improved activity. (S)-RET titers and final $OD_{600}$ in engineered strains carrying

**b** Various combinations of NMCH and CPR, and **c** Multiple copies of rate-limiting enzymes. Significance was calculated using two-tailed $t$-test. Data are presented as mean ± standard deviations ($n = 4$ biologically independent samples). Source data are provided as a Source Data file.

optimal enzyme, four BBEs including PsBBE from *P. somniferum*, CjBBE from *C. japonica*, CyBBE from *C. yanhusuo*, StBBE from *Stephania tetrandra S. Moore* were compared in a low (S)-RET producing strain XJ048 (Supplementary Fig. 12). Expressing all candidates except for StBBE led to (S)-SCO production, with CyBBE showing the highest activity. However, even in the CyBBE expressing strain the remaining amounts of (S)-RET was approximately twice of the produced (S)-SCO in cells (Supplementary Fig. 12), indicating a potential for further improvement on (S)-RET conversion efficiency.

A previous in vitro study reported the optimum pH at 8.9 for BBE[48], while in vivo experiment has revealed that BBE, containing an ER signal peptide and a vacuolar sorting signal, was finally localized to the vacuole in plant, where the pH is supposed to be acidic[22]. We hypothesized this divergence resulted in the low activity of BBE. To compromise such issue, we sought to retarget such enzyme into various subcompartments. In yeast, pH in the mitochondrial matrix was reported to be 7.0-7.5, cytosol and ER pH both around 7.0, and pH in the Golgi network gradually decreases from *cis*-Golgi to *trans*-Golgi (from 6.5 to 6.0)[49]. To express CyBBE in the cytosol, we first attempted N-terminal truncations to generate two variants CyBBEΔ29 and CyBBEΔ46, by removal of predicted ER targeting signal, or both ER targeting signal and vacuolar sorting signal, respectively. Besides, additional truncation CyBBE29Δ46 was also constructed by deleting vacuole signal but retaining ER signal (Supplementary Fig. 13a and d). Unexpectedly, all three CyBBE variants resulted in undetectable (S)-SCO production (Supplementary Fig. 13b), indicating that such truncations affected its proper expression or correct folding. Indeed, fluorescence analysis showed that CyBBEΔ29 and CyBBEΔ46 cytosolically expressed but along with severe aggregate formation, while

CyBBE29Δ46 might enter the trafficking pathway due to the surviving ER signal peptide (Supplementary Note 1 and Supplementary Fig. 13f). Furthermore, western blot revealed that wild type CyBBE and CyBBE29Δ46 were blotted with larger size bands than that of CyBBEΔ29 and CyBBEΔ46, both of which expressed similarly as expected (Supplementary Note 1 and Supplementary Fig. 13c). Interestingly, after the treatment of PNGaseF, capable of specifically removing the N-linked glycosylation, the size shift was no longer observed for the wild type CyBBE and CyBBE29Δ46 (Supplementary Fig. 13e). These results clearly suggested that CyBBE underwent trafficking and post-translational modifications, such as N-linked glycosylation.

Given the importance of both signals for CyBBE, we turned to retrograde CyBBE from Golgi to ER or retain it in Golgi, instead of trafficking the enzyme to the native destination vacuole. We tailored CyBBE with an ER C-terminal HDEL peptides (CyBBE_ERTS) to mimic ER lumen soluble protein[50], or a N-terminal 118-amino acid Golgi targeting sequences (GOTS_CyBBE) to mimic type II Golgi integral membrane protein[51], and tested two variants in the (S)-RET platform strain XJ0691. GOTS_CyBBE and CyBBE_ERTS led to the improvement of (S)-SCO production by 33% ($p = 0.0001$) and 206% ($p < 0.0001$), respectively, compared to the wild type CyBBE (Fig. 4b and Supplementary Fig. 15a). To determine where both variants and wild type CyBBE localized in yeast, we performed colocalization microscopy analysis of green fluorescent protein (GFP)-tagged both variants and wild type CyBBE, and red fluorescent protein mRuBy2-fused marker proteins for ER, vacuole, and Golgi (*Elo3*, *Vph1* and *Chs5*, respectively)[52]. The results confirmed that wild type CyBBE was transported to the vacuole, GOTS_CyBBE enabled retention in the Golgi, while CyBBE_ERTS was redirected back to the ER (Fig. 4a and b). To better understand possible

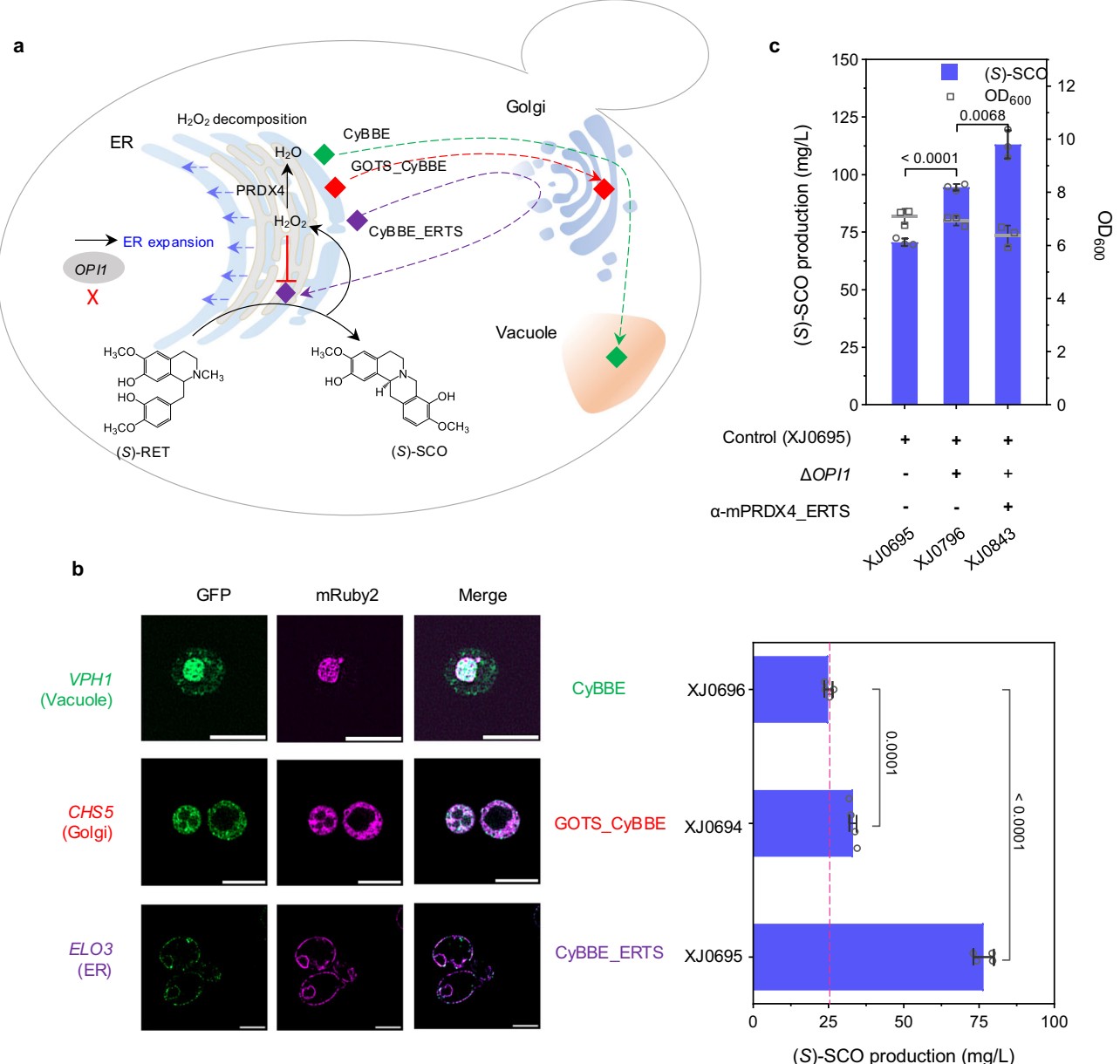

**Fig. 4 | Fine-tuning BBE to improve (S)-SCO production. a** Schematic illustration of module III regarding fine-tuning BBE to increase (S)-SCO production. Green diamond indicates wild type CyBBE, transporting to vacuole; red diamond indicates Golgi-retained GOTS_CyBBE, retaining in Golgi; purple diamond indicates ER-targeted CyBBE_ERTS, retrograding to ER. **b** Fluorescence colocalization of CyBBE and its variants, (S)-SCO titers and final $OD_{600}$ in engineered strains, in which CyBBE was tailored to target various organelles. Scale bar represents 5 μm. **c** (S)-SCO titers and final $OD_{600}$ obtained from the engineered strains, in which ER was expanded by *OPI1* deletion, and ER-localized mPRDX4 was expressed to alleviate the stress induced by $H_2O_2$ production. Significance was calculated using two-tailed *t*-test. Data are presented as mean ± standard deviations (*n* = 3 or 4 biologically independent samples). Source data are provided as a Source Data file.

mechanisms of ER retrograding strategy, we performed RT-qPCR, western blot, in vitro activity assays with various pH, as well as molecular dynamic simulations to compare the difference between CyBBE_ERTS and wild type CyBBE (Supplementary Note 2). Although GOTS_CyBBE expression was significantly increased, CyBBE_ERTS expression exhibited a comparable level to the wild type CyBBE (Supplementary Fig. 15b and c), indicating that over two-fold increase of product (S)-SCO was caused by increased enzyme activity rather than expression level of CyBBE_ERTS. Also, western blot with the treatment of PNGaseF for CyBBE_ERTS showed a similar pattern as the wild type CyBBE (Supplementary Fig. 15d), suggesting that they have gone through similar post-modifications. Moreover, in vitro assays confirmed that higher pH (as comparing the environment in ER to vacuole) indeed favored the efficiency of BBE-catalyzed reaction,

which matched well with the proposed molecular mechanisms from enzyme and substrate level (Supplementary Note 2 and Supplementary Fig. 16a–f). Molecular dynamic simulation inferred that the C-terminus HDEL was beneficial to the stability of CyBBE (Supplementary Fig. 17a–f). Therefore, such ER targeting strategy guaranteed CyBBE going through secretory pathway for proper processing and post-modifications, while at the same time the strategy could retrograde it to ER, which provides a more favorable microenvironment thereby leading to its increased activity.

While CyBBE_ERTS promoted (S)-SCO titer up to 76.5 mg/L (strain XJ0695), more than 100 mg/L substrate (S)-RET remained (Supplementary Fig. 15a), suggesting limitations persisted for this step. We initially questioned if substrate availability was the issue since (S)-RET was detected extracellularly. We thus expressed two transporters

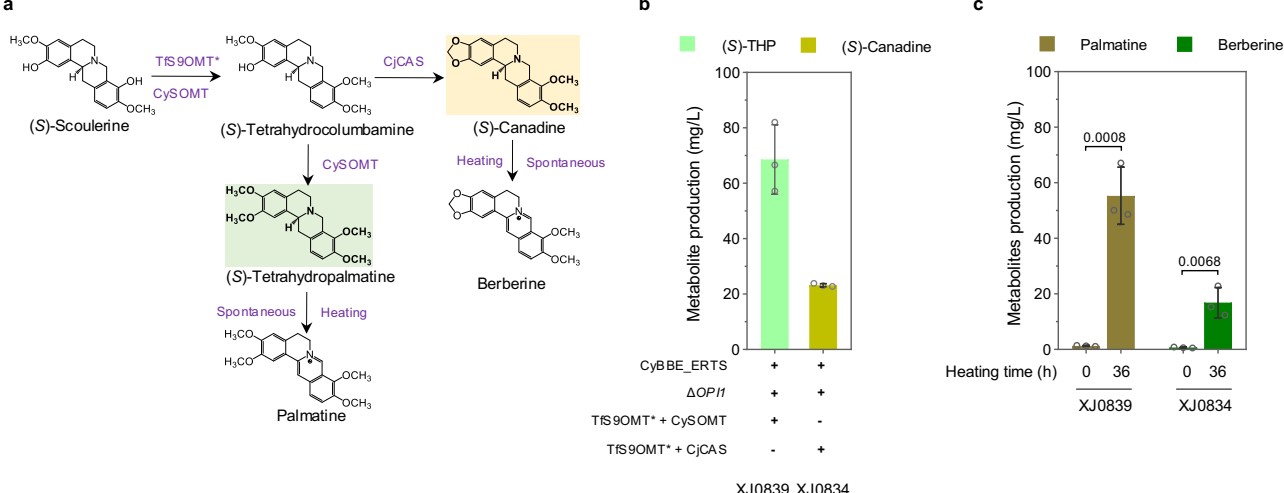

**Fig. 5 | Extending the pathway to biosynthesize palmatine in module IV and berberine in module V. a** Schematic presentation of module IV and module V. **b** (*S*)-THP and (*S*)-canadine titers in strains XJ0839 and XJ0834, respectively. **c** 36h-heating of fermentation broth led to increased palmatine and berberine titers.

Significance was calculated using two-tailed *t*-test. Data are presented as mean ± standard deviations (*n* = 3 biologically independent samples). Source data are provided as a Source Data file.

PsPUB1 and PsBUP6, derived from *P. somniferum*[53], to enhance the re-uptake efficiency of extracellular (*S*)-RET. However, no significant effects on (*S*)-SCO titers were observed (Supplementary Fig. 18a). Next, we attempted to spatially express Ps4′OMT in close proximity to ER-targeted CyBBE_ERTS by localizing Ps4′OMT to ER membrane via fusion with a N-terminal ER membrane-bound *CYB5* (CYB5-Ps4′OMT), or to the lumen with a N-terminal SP$_{CyBBE}$ and C-terminal ERTS (SP$_{CyBBE}$-Ps4′OMT_ERTS), or targeting the fusion of both enzymes to ER lumen (SP$_{CyBBE}$-Ps4′OMT-CyBBE_ERTS, CyBBE-Ps4′OM-T_ERTS). Direct fusions resulted in a dramatic decrease on (*S*)-SCO production whereas CYB5-Ps4′OMT and SP$_{CyBBE}$-Ps4′OMT_ERTS did not affect (*S*)-SCO production (Supplementary Fig. 18b). These results suggested that substrate availability was not a limiting factor at the current stage.

Secondly, we suspected that excess substrate (*S*)-RET remaining was likely due to the inefficiency of the CyBBE_ERTS variant. However, extra copy introduction of wild type CyBBE, GOTS_CyBBE or CyBBE_ERTS in strain XJ0695 resulted in a varying decrease on the (*S*)-SCO titers (Supplementary Fig. 18c), indicating other factors limiting this step.

Thirdly, we considered the potential toxicity of the products, (*S*)-SCO and hydrogen peroxide ($H_2O_2$), as both have been shown to inhibit BBE activity[48]. Since (*S*)-SCO was mainly detected extra-cellularly, our attention was turned to $H_2O_2$. We expressed ER-localized catalase *Ctt1*, *Cta1* and one *Cta1* mutant (removal of potential N-glycosylation N241), through fusion with N-terminal signal peptide SP$_{kar2}$ and C-terminal ERTS in strain XJ0695. All candidates caused a retarded growth and around 40% (*p* < 0.0001) decrease on the (*S*)-SCO titers (Supplementary Fig. 19a). Instead of troubleshooting on these constructs, we tested a mammalian based peroxiredoxin IV (PRDX4) with a N-terminal signal sequence and a C-terminal ERTS (α-mPRDX4_ERTS)[54], which could transfer the electrons derived from protein oxidative folding to $H_2O_2$, thereby producing $H_2O$ (Fig. 4a and Supplementary Fig. 19b). As a control, two nonsense mutants α-mPRDX4$^{C127S}$_ERTS and α-mPRDX4$^{C248S}$_ERTS were also generated by mutating either of active-site cysteines. Intriguingly, α-mPRDX4_ERTS expression improved (*S*)-SCO titer up to 98.4 mg/L, 30% (*p* = 0.0028) more than that in control strain XJ0695, whereas both mutants led to decrease on (*S*)-SCO titer (Supplementary Fig. 19b). These results confirmed the toxicity of $H_2O_2$ was a potential issue for CyBBE_ERTS activity.

Fourthly, since BBE is a FAD-dependent oxidase which undergoes post-translational modification through the trafficking pathway[55], we thought to engineer trafficking pathway as an alternative to optimize CyBBE expression through 1) ER expansion, by deletion of *PAH1*, encoding $Mg^{2+}$ - dependent phosphatidate (PA) phosphatase which could dephosphorylate PA to yield diacylglycerol, *OPI1*, encoding a transcriptional regulator which negatively regulates phospholipid biosynthesis, or overexpression of *INO2*, encoding a transcription activator for phospholipid biosynthesis, INO2* (encoding an Opi1-insensitive Ino2$^{L119A}$)[56]; 2) Protein folding, by overexpression of sliced *HAC1*, chaperone *CPR5* or deletion of kinase *GCN2*; 3) Impairing the ER associated degradation via *HRD1* or *DER1* deletion to minimize aberrant protein retro-translocation to the cytoplasm and offer extra time for correctly folding, or disruption of the vacuolar proteinase *PEP4* to reduce protein degradation; 4) Overexpression of HDEL receptors *ERD1* or *ERD2* to improve the retention of HDEL-carrying protein in ER by alleviating the readily saturated receptor-mediated retrieval of luminal ER proteins from the secretory pathway. Among all tested targets, only overexpression of *INO2*, INO2*, deletion of *OPI1* improved (*S*)-SCO production by 17% (*p* = 0.4225), 18% (*p* = 0.1726), and 36% (*p* = 0.0022), respectively, whereas overexpression of *CPR5* and dele-tion of *HRD1*, *DER1* exhibited a severely opposite impact (Supplementary Fig. 19c). Taken together, we first improved BBE activity by ER compartmentalization strategy, and observed that *OPI1* deletion-mediated ER expansion caused extra 36% (*p* < 0.0001) increase on (*S*)-RET conversion, and further alleviating the feedback inhibition of by-product $H_2O_2$ via expression of mammalian-derived PRDX4 resul-ted in additional 30% (*p* = 0.0068) increase on (*S*)-RET conversion. With the combination of all these strategies, the final strain XJ0843 produced 113.1 mg/L (*S*)-SCO, representing the highest titer ever reported. (Fig. 4c).

## De novo production of protoberberine alkaloids in yeast

In modules IV and V, we aimed to biosynthesize two protoberberine alkaloids, palmatine and berberine (Fig. 1). Palmatine biosynthesis involved three steps from (*S*)-SCO, which are catalyzed by scoulerine 9-*O*-methyltransferase (S9OMT), columbamine *O*-methyltransferase (CoOMT) and tetrahydroprotoberberine oxidase (Fig. 5a and Supple-mentary Fig. 20a). Initially, we sought to select the optimal methyl-transferases from TfS9OMT* (*Thalictrum flavum*)[24], CjCoOMT (*C. japonica*), ThCoOMT (*Thalictrum thalictroides*) and CySOMT

(*C. yanhusuo*) in genetic-background-simple strain XJ083. Apparently, a bifunctional methyltransferase CySOMT, converting (*S*)-SCO to (*S*)-tetrahydropalmatine ((*S*)-THP) directly, exhibited a lower activity on converting (*S*)-SCO to (*S*)-tetrahydrocolumbamine ((*S*)-THC) than TfS9OMT*, and a comparable activity on (*S*)-THP production to CjCoOMT (Supplementary Fig. 20b). We then transformed TfS9OMT* and CySOMT, combining with GOTS_CyBBE, CyBBE_ERTS, or wild type CyBBE, respectively, into (*S*)-RET producing platform strain XJ0691. The derived strain XJ0839 containing CyBBE_ERTS, TfS9OMT*, CySOMT and the deletion of *OPI1* produced the highest (*S*)-THP with 68.6 mg/L (Fig. 5b and Supplementary Fig. 20c and f). Palmatine is biosynthesized from the oxidation of (*S*)-THP by losing four electrons. Currently, there are a few plant-derived flavoprotein oxidases reported to catalyze this type of reactions in vitro, such as STOX and DBOX candidates, but their heterologous function remained unexplored in yeast. Six codon-optimized STOX homologs, including AmSTOX from *Argemone Mexicana*[25], BwSTOX from *Berberis wilsoniae*[25], CjTHBO from *C. japonica*[26], PsFADOX5 from *P. somniferum*[27], McDBOX1, McDBOX2 from *M. cordata*[28] and four fusions, HPA2-AmSTOX (HPA2 encoding tetrameric histone acetyltransferase), CySOMT-AmSTOX, HPA2-PsFADOX5, CySOMT-PsFADOX5, were transformed into strain XJ0700 (XJ0691+CyBBE_ERTS+TfS9OMT*+CySOMT). All resultant strains and the control strain XJ0700, produced comparable palmatine at low quantities, which might be from spontaneous, instead of enzymatic reaction (Supplementary Fig. 20d and e). Nevertheless, we observed that the production levels of (*S*)-THC and (*S*)-THP in strains XJ0727 (expressing McDBOX1), XJ0728 (expressing McDBOX2) and XJ0841 (expressing McDBOX2) significantly decreased compared to their control strains XJ0700 or XJ0839 (Supplementary Fig. 20e and f), indicating that some unknown compounds were produced from (*S*)-THC and (*S*)-THP by the expression of McDBOX1 or McDBOX2. Intermediates analysis of stain XJ0841 expressing McDBOX2 by LC-MS/MS showed that a significant peak of $m/z$ 354.16 [M]$^+$, which exhibited different retention time (26.93 min) and tandem mass spectrum (MS2) with that of commercial standard 7,8-dihydropalmatine (27.32 min) bearing the same $m/z$ (Supplementary Fig. 21b–d). Moreover, we found that the standard 7,8-dihydropalmatine was highly unstable, which would be totally transformed into palmatine even in minimal media during 24 h, whereas the produced compounds remained stable in shake flask fermentation even after 144 h (Supplementary Fig. 22a and b). Previous studies have confirmed that the oxidation of tetrahydroprotoberberine catalyzed by STOX only occurred to its C-ring[27], and the oxidative product with N7 = C14 iminium bond was unstable, undergoing rapid disproportionation to form a mixture of the corresponding quaternary protoberberine and the tetrahydroprotoberberine[57]. Therefore, we inferred that (*S*)-THP was oxidized to produce 13,14-dihydropalmatine harboring N7 = C8 bond by expressing McDBOX2 in strain XJ0841 (Supplementary Fig. 21a), as the production of quaternary palmatine was not increased (Supplementary Fig. 20f). Similarly, (*S*)-THC was transformed into 13,14-dihydrocolumbamine by McDBOX2 oxidation in such strain (Supplementary Fig 21e and f), and (*S*)-SCO was transformed into 13,14-dihydroscoulerine in strain XJ0842 with McDBOX2 expression (Supplementary Fig. 21g and h). Recently, one study reported that heating enhanced the transformation from tetrahydroprotoberberine to the corresponding quaternary protoberberine via losing four electrons[58]. We therefore tested heating the cultures from shake flask fermentation of strain XJ0839 at 98 °C for varying durations. After heating for four hours, 72% ($p = 0.0034$) (*S*)-THP (49.2 mg/L) disappeared, while 14.3 mg/L palmatine was produced (Supplementary Fig. 24a). By increasing heating time to 36 hours, 91% ($p < 0.0001$) (*S*)-THP was transformed into 54.0 mg/L palmatine with 86% ($p = 0.0008$) conversion efficiency (Fig. 5c and Supplementary Fig. 24a).

In parallel, berberine biosynthesis undergoes three steps from (*S*)-SCO, catalyzed by S9OMT, canadine synthase (CAS) and tetrahydroprotoberberine oxidase (Fig. 5a and Supplementary Fig. 23a). With the introduction of TfS9OMT* and CjCAS (*C. japonica*) into strain XJ0796 (XJ0695 *ΔOPI1*), its resultant strain XJ0834 produced 23.2 mg/L (*S*)-canadine (Fig. 5b and Supplementary Fig. 23c). We further attempted to express CjTHBO, ER-targeted CjTHBO_ERTS, Golgi-targeted GOTS_CjTHBO or McDBOX2 in strain XJ0834. Apparently, all four enzymes exhibited no activity on berberine production improvement (Supplementary Figs. 23b and c). Only the expression of McDBOX2 in strain XJ0835 (XJ0834 + McDBOX2) allowed for the conversion of (*S*)-THC and (*S*)-canadine to 13,14-dihydrocolumbamine and 13,14-dihydroberberine, respectively (Supplementary Fig. 23d–f). Furthermore, we sought to convert (*S*)-canadine to berberine through heating for varying durations at 98 °C. Heating the cultures of strain XJ0834 by four hours led to that 87% ($p < 0.0001$) (*S*)-canadine was transformed while berberine production was only 3.9 mg/L. Once prolonging the heating time until 36 hours, berberine production was gradually increased, reaching up to 16.9 mg/L finally (Fig. 5c and Supplementary Fig. 24c). It is worth noting that 13,14-dihydropalmatine and 13,14-dihydroberberine from strains XJ0841 and XJ0835, respectively, were readily oxidized by heating (Supplementary Fig. 24b and d). In addition, another two types of protoberberines, dehydroscoulerine and columbamine were also detected after four hours' heating (Supplementary Fig. 25a–f).

## De novo production of benzophenanthridine alkaloids in yeast

The module VI was to extend the pathway for production of sanguinarine from (*S*)-SCO, sequentially catalyzed by cheilanthifoline synthase (CFS), stylopine synthase (SPS), tetrahydroprotoberberine *N*-methyltransferase (TNMT), methylstylopine 14-hydroxylase (MSH), protopine 6-hydroxylase (P6H) and DBOX (Fig. 1). To reconstruct the module VI more accessible, we first extended the downstream pathway to produce the intermediate protopine (Supplementary Fig. 28a). Two enzyme candidates from *E. californica* and *C. yanhusuo* for first two steps were co-expressed with PsTNMT and PsMSH (both from *P. somniferum*) in (*S*)-SCO-producing strain XJ0695. Total ion chromatography (TIC) and MS2 analysis evidenced that the intermediate protopine was produced, and the enzyme combination of EcCFS and EcSPS exhibited the optimal activity (Supplementary Fig. 28b). Prior studies have reported trace amounts of sanguinarine production from dihydrosanguinarine in yeast, by spontaneous conversion or unknown yeast native enzymatic catalyzation[18,19]. As aforementioned, this kind of reaction can be catalyzed by flavoprotein oxidases. Thus, six previously mentioned STOX homologs, AmSTOX, BwSTOX, CjTHBO, PsFADOX5, McDBOX1 and McDBOX2 were transformed together with EcP6H (from *E. californica*) into protopine-producing strain XJ0743. Western blot analysis revealed that BwSTOX, McDBOX1 and McDBOX2 expressed well, PsFADOX5 smeared, whereas AmSTOX and CjTHBO were not properly expressed (Supplementary Fig. 29a and b). However, we did observe that all transformants produced sanguinarine even in the control strain XJ07437 only expressing EcP6H, consistent with previous observations of low sanguinarine production without functional oxidase catalyzation in yeast[18,19] (Supplementary Fig. 28c and e). Surprisingly, the dihydrosanguinarine peak disappeared from strain XJ0750 expressing McDBOX2 (Supplementary Fig. 28c), inferring that McDBOX2 was functional in conversion of dihydrosanguinarine to sanguinarine. With the introduction of McDBOX2, combined with EcP6H, McP6H, or CyP6H, all resultant strains could produce sanguinarine with EcP6H exhibiting the optimal performance in strain XJ0750 (Supplementary Fig. 28d).

Sanguinarine downstream derivative chelirubine, exhibiting the anti-proliferative effects on several cancer cell lines can be used as a DNA fluorescent probe, which have been drawn increased interest[14,59]. It requires three enzymes, containing P450 hydroxylase, *O*-methyltransferase and flavoprotein oxidase to fulfill chelirubine biosynthesis from dihydrosanguinarine as illustrated in module VII (Fig. 1 and

Supplementary Fig. 31a). We first expressed three P450s CYP82P2, CYP82P3 and CYP82P4 from *E. californica* in sanguinarine producing strain XJ0822 containing McDBOX2. A peak corresponding to *m/z* 348.09 [M]$^+$ was observed only in strain XJ0851 expressing CYP82P2 (Supplementary Fig. 31b and c), presumably indicating that CYP82P2 catalyzed dihydrosanguinarine to generate 10-hydroxydihydrosanguinarine, which was then oxidized to generate 10-hydroxysanguinarine by McDBOX2 catalyzation. Subsequently, we characterized two potential *O*-methyltransferases Ec2OMT and Ec11OMT from *E. californica* together with CYP82P2 in strain XJ0832. A peak corresponding to *m/z* 362.1 [M]$^+$ was observed only in strain expressing CYP82P2 and Ec11OMT, the MS2 spectrum of which peak was mostly identical to that of chelirubine standard reported in a previous study[60] (Supplementary Fig. 31d and e). All these results demonstrated that chelirubine was de novo biosynthesized in yeast.

Likewise, we extended the pathway to generate chelerythrine from (*S*)-SCO in module VIII by further introduction of TfS9OMT*, CAS, PsTNMT, PsMSH, EcP6H and McDBOX2 (Fig. 1). To produce intermediate allocryptopine, CjCAS from *C. japonica* or CyCAS from *C. yanhusuo*, TfS9OMT*, PsTNMT and PsMSH were chromosomally incorporated into (*S*)-SCO producing strain XJ0695 (Supplementary Fig. 32a). Both enzyme combinations enabled production of allocryptopine, while CjCAS in XJ0741 exhibited a higher activity compared to CyCAS in strain XJ0742 (Supplementary Fig. 32b). With the introduction of McDBOX2, combined with EcP6H, McP6H, or CyP6H, all resultant strains produced chelerythrine with EcP6H performing the best in strain XJ0747 (Supplementary Fig. 32c). These results indicated that McDBOX2 exhibited a broad substrate scope, leading to the biosynthesis of chelerythrine, sanguinarine and chelirubine.

### Optimizing DBOX expression for the improvement of BZDAs production

McDBOX2 could catalyze various substrates, playing a vital role in BZDAs biosynthesis (Fig. 6a). However, we observed that McDBOX2 expression caused a decreased OD$_{600}$ and an increased (*S*)-RET accumulation in both chelerythrine producing strain XJ07411 and sanguinarine producing strain XJ07437 (Fig. 6b). Likewise, decreased (*S*)-SCO production and increased (*S*)-RET accumulation were also found in (*S*)-SCO producing strain XJ0695 when expressing McDBOX2 (Fig. 6b). These results indicated that McDBOX2 expression displayed some limitations on BBE activity, which is unexpected as CyBBE was engineered to localize inside ER, while wild type McDBOX2 was assumed to be in vacuole in our case (Fig. 4b, d). Although they are not spatially localized together, there are three shared traits between CyBBE_ERTS and McDBOX2 expression in yeast, including being FAD-dependent oxidases, producing by-product H$_2$O$_2$, and undergoing trafficking pathway from ER to Golgi. H$_2$O$_2$ production was not considered since H$_2$O$_2$ generated by McDBOX2 was in vacuole which might not affect ER-localized CyBBE. Then we considered the potential competition for the cofactor FAD. We sought to improve FAD availability by transforming plasmids 1) bearing FAD ER-localized transporters *FLC1*, *FLC2*, *FLC3* and *YPR365C*; 2) mitochondria FAD transporter *FLX1*; 3) plasma transporter *MCH5*; 4) FAD biosynthesis-related genes *FMN1* and *FAD1*; 5) BsRiba and BsRibc from *Bacillus subtilis* into strain XJ0774 (Supplementary Fig. 34a). We observed that the derived strains expressing MCH5 or BsRibc exhibited 24% (*p* = 0.0095) and 39% (*p* = 0.0038) increase on chelerythrine production, respectively, compared to the control strain expressing empty plasmid, while others displayed no significant impacts on chelerythrine production (Supplementary Fig. 34b). However, comparable (*S*)-RET titer remained in strains expressing MCH5, BsRibc or empty plasmid (Supplementary Fig. 34b), which indicated that MCH5 or BsRibc expression probably improved McDBOX2 activity at this stage thereby leading to increased chelerythrine production, instead of alleviating the limitation on CyBBE_ERTS for conversion of (*S*)-RET to (*S*)-SCO. Next, we engineered

the coat protein II (COPII) vesicles for transporting from ER to Golgi in strain XJ0819 by plasmid-based overexpression of *SAR1* or replacement of the promoter *SEC16* with strong promoters *GPD1*p or *TEF1*p. *SAR1* overexpression resulted in severe growth retardation and decrease on chelerythrine titer, whereas promoter substitution of *SEC16* showed no significant effect on chelerythrine production compared to the control strain (Supplementary Fig. 35).

Although the mechanism of such limitation is unknown, we thought to circumvent the issue via different compartmentalization of McDBOX2. Since most vacuole proteins suffer from degradation in the vacuole, we first attempted to target McDBOX2 into the ER. Western blot confirmed that McDBOX2_ERTS expressed well in the ER, identical to the wild type enzyme (Supplementary Fig. 36a). However, ER-targeted McDBOX2 didn't alleviate the issue as (*S*)-RET titers and final OD$_{600}$ were comparable with that of the strains expressing vacuolar McDBOX2, even slightly decreased chelerythrine and sanguinarine production (Supplementary Fig. 36b). Next, cytosolic expression of McDBOX2 was tested by truncating its N-terminal 29 amino acids. Western blot analysis suggested that one protein band appeared from the strain expressing McDBOX2 truncation, the size of which was close to the theoretical 56.8 kDa of McDBOX2Δ29, whereas wild type enzyme was blotted with several bands (Supplementary Fig. 36c). We observed that McDBOX2Δ29, which was cytosolically expressed as demonstrated by fluorescent tagging, resulted in unarresting cell growth and parallel (*S*)-RET titers, compared to those of strain XJ07411 without expressing McDBOX2 (Fig. 6c–e). Furthermore, such truncation in strain XJ07472 (XJ07411 + McDBOX2Δ29) could convert all the dihydrochelerythrine to chelerythrine, the production of which was, however, slightly less than that of strain XJ0747 (XJ07411 + McDBOX2) after 40 hours' growth, probably due to more accumulated intermediates, (*S*)-THC and (*S*)-*cis*-*N*-methylcanadine, instead of decreased activity of McDBOX2Δ29 (Fig. 6c, e and Supplementary Fig. 37a). Likewise, McDBOX2Δ29 in strain XJ07502 (XJ07437 + McDBOX2Δ29) resulted in recovering growth, accumulating less (*S*)-RET, converting all dihydrosanguinarine, and producing more sanguinarine, compared to those of the control strain XJ0750 (Supplementary Fgs. 36d, e and b). It suggested that the functional McDBOX2Δ29 to some extent alleviated the limitation on BBE. To improve chelerythrine and sanguinarine production, we leveraged the aforementioned strategies to engineer the strain XJ07472 and XJ07502, both expressing McDBOX2Δ29. In our final strains XJ07474 and XJ07504, combination of riboflavin transporter *MCH5*, FAD synthase BsRibc and ER expansion mediated by *OPI1* deletion resulted in 38.1 mg/L of chelerythrine, and 3.8 mg/L of sanguinarine, respectively (Fig. 6f and Supplementary Fig. 36f). Moreover, intracellular product levels were measured in strains XJ0827 and XJ0832 by using ACN extraction. 9.6% (*p* = 0.0008) chelerythrine and 22% (*p* < 0.0001) sanguinarine were accumulated in the cell pellet, while less than 6% (*p* = 0.0002) of other intermediates were observed intracellularly, indicating that compared to final product chelerythrine and sanguinarine, other intermediates were more readily secreted into medium (Supplementary Table 1).

## Discussion

In this study, we demonstrated de novo biosynthesis of a series of protoberberines, including columbamine, palmatine, berberine, and BZDAs, including sanguinarine, chelerythrine and chelirubine in yeast. Initially, we constructed the distinct platform strains for producing (*S*)-NOR (XJ0636 315.9 mg/L), (*S*)-RET (XJ0691 425.2 mg/L) and (*S*)-SCO (XJ0843 113.1 mg/L), respectively, through a combination of various strategies, including enlarging the tyrosine pool, impairing the undesirable 4-HPAA scattering, increasing copies of rate-limiting enzymes, introducing plant-derived tyrosine pathway, 4-HPAA biosynthesis pathway, CPR pairing, BBE relocalization, ER expansion and PRDX4-mediated H$_2$O$_2$ degradation. Subsequently, the (*S*)-SCO platform strain was engineered to produce tetrahydroprotoberberines and

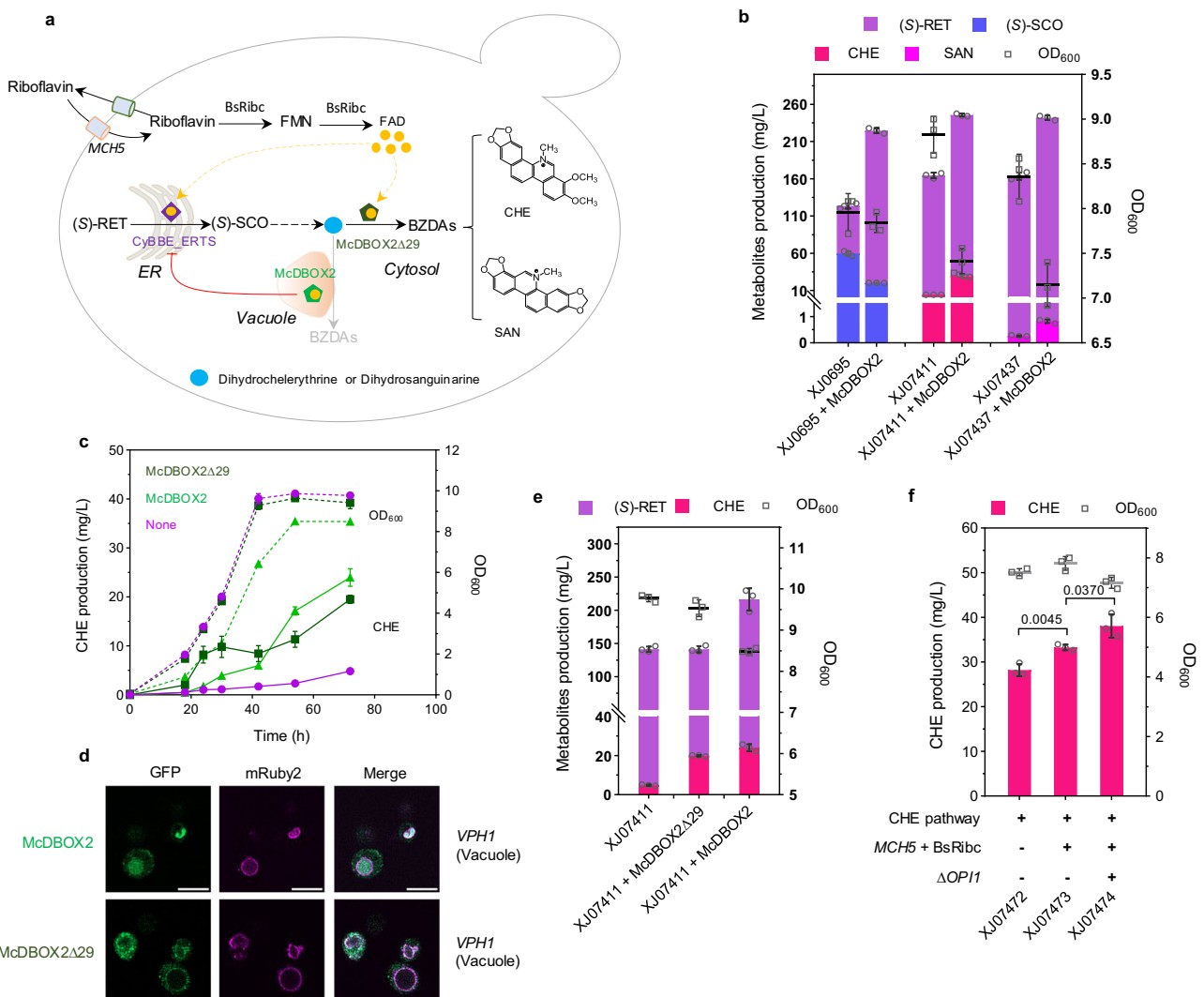

**Fig. 6 | Optimizing the biosynthesis of BZDAs in yeast. a** Schematic representation of strategies for chelerythrine (CHE) and sanguinarine (SAN) production. *MCHS*, plasma-bounded riboflavin transporter; BsRibc, FAD synthase from *B. subtilis*. Orange circle indicates FAD; purple hexagon indicates ER-targeted CyBBE_ERTS; green and dark green hexagon indicate vacuole-localized McDBOX2 and cytosolically expressed McDBOX2Δ29, respectively. **b** Metabolite titers and final OD$_{600}$ in engineered strains with McDBOX2 expression. **c** Time course of CHE titer and OD$_{600}$ in engineered strains expressing McDBOX2Δ29, McDBOX2 or no DBOX candidate. The solid line represents CHE production; the dashed line represents OD$_{600}$. Purple indicates the strain XJ07411 without the expression of

DBOX candidate; light green indicates the strain XJ0747 (XJ07411 + McDBOX2) expressing wild type McDBOX2; dark green indicates the strain XJ07472 (XJ07411 + McDBOX2Δ29) expressing McDBOX2Δ29. **d** Fluorescence analysis of McDBOX2Δ29 and McDBOX2 localization in yeast. Scale bar represents 5 μm. **e** (*S*)-RET, CHE titers and OD$_{600}$ in engineered strains XJ07411, XJ0747 (XJ07411 + McDBOX2) and XJ07472 (XJ07411 + McDBOX2Δ29). **f** Increased CHE production by improving FAD availability and *OPI1* deletion-mediated ER expansion. Significance was calculated using two-tailed *t*-test. Data are presented as mean ± standard deviations (n = 3 or 4 biologically independent samples). Source data are provided as a Source Data file.

dihydrobenzophenanthridine alkaloids, which can be converted into 13,14-dihydroprotoberberines and BZDAs by FAD-dependent oxidase McDBOX2. Meanwhile, we observed that tetrahydroprotoberberines and 13,14-dihydroprotoberberines can be transformed into protoberberines by heating or spontaneously, and cytosolically expressed McDBOX2 could alleviate some limitations on CyBBE_ERTS activity. These results highlight the potential of yeast cell factories to access and scall up the production of distinct valuable BIAs for drug development.

The nascent BBE protein matures via the secretory pathway and is ultimately localized to plant vacuole[21,22]. In this study, we have confirmed that CyBBE was functionally expressed and finally targeted to the vacuole in yeast, the activity of which on (*S*)-RET, however, was quite low (Fig. 4b and supplementary Fig. 12a). Previous attempts to improve the efficiency of BBE, such as multiple chromosomal integration and BBE truncations, didn't yield significant improvement[20,23].

Recent studies have reported that replacing the N-terminal signal peptides of THCAS and CBDAS with a vacuolar localization tag from yeast, and engineering AbLS by N-terminal fusions which may alter sorting from the *trans*-Golgi network TGN, enabled these plant-based vacuolar proteins functionally expressed in yeast[61,62]. Our initial attempts, including signal peptides replacement of CyBBE with a signal sequence from vacuolar proteinase A and engineering CyBBE by N-terminally fusing soluble domains with various oligomerization state, resulted in no significant impact on (*S*)-SCO titers (Supplementary Fig. 14a and b). Given that each subcellular compartment in yeast offers a unique physiochemical environment, we questioned if the activity of CyBBE was improved when expressed in various compartments in yeast, including peroxisome, ER, Golgi and cytosol. The peroxisomal variant CyBBE_ePTS1, led to comparable (*S*)-SCO production to wild type CyBBE (Supplementary Fig. 14a), while cytosolically expressed truncations CyBBEΔ29 and CyBBEΔ46 almost

completely lost their activity (Supplementary Fig. 13b), suggesting the importance of guiding CyBBE into secretory pathway for proper expression or correct folding (Supplementary note 1).

Regarding the secretory pathway in yeast, it shares most organelles and post-transcriptional modifications within plants, and is responsible for the synthesis, folding and delivery of a diverse array of cellular proteins among these organelles. Guided by signals, these proteins first enter the ER, further transport to the Golgi to ensure proper folding and modification. Afterwards, some proteins are retrograded to ER, while others reside in Golgi or are transported to later compartments, such as the vacuole and extracellular space. For instance, the proteins with vacuole sorting sequences will go from *trans*-Golgi apparatus to the vacuole, including endogenous Pep4p and BBE in this study. In addition, there are some proteins that first reach the *cis*-Golgi apparatus and then retrieve back to the ER for proper function. These protein includes 1) the soluble ER proteins, carrying a C-terminal sequence HDEL[50,63], such as Kar2p; 2) the type I integral membrane ER proteins, possessing an important C-terminal cytoplasmic motif[64,65], such as Wbp1p; and 3) type II integral membrane ER proteins, bearing a N-terminal cytoplasmic ER retention signal[66], such as Mns1p and Alg5p. Currently, the commonly used strategy of ER localization in metabolic engineering was to mimic the type II integral membrane ER proteins by signal replacement, as demonstrated in one recent study of targeting CjSTOX into ER via Mns1 signal peptide substitution[58]. In our case, we have mimicked both type II transmembrane ER protein $SP_{ALG5}$_CyBBE though N-terminal Alg5p signal peptide replacement and type I transmembrane ER protein CyBBE_$C36_{WBP1}$ though C-terminal fusion with the last 36 amino acids of Wbp1p, both of which, however, almost lost their activity on (S)-SCO production (Supplementary Fig. 15e and f). Intriguingly, we verified that ER offered a more favorable environment for CyBBE activity by mimicking soluble ER protein CyBBE_ERTS though C-terminal HDEL fusion (Supplementary note 2). This ER retrograding strategy would be generally applicable for heterologous expression of specific enzymes requiring post-translational processing. In addition, each compartment has their unique physicochemical environment, such as pH and oxidative status, which may provide favorable conditions for the function of distinct enzymes. In yeast, it is known that vacuolar has a more acidic environment while the pH in ER is close to neutral. We now have shown that pH is an important factor rendering ER-localized BBE outperforming the vacuolar targeted BBE (Supplementary note 2 and Supplementary Fig. 16a–f). Therefore, this ER retrograding strategy provides a more favorable microenvironment (i.e., pH) for maximizing the catalytic activity of BBE, which could be applicable for expression of enzymes requiring proper pH conditions. Moreover, the C-terminal cytosolic tail results in Golgi retention of type I integral membrane proteins, Kex1p and Kex2p[67,68], while the N-terminal cytoplasmic domain containing aromatic residues (FXFXD) is both necessary and sufficient for Golgi retention of type II integral membrane protein Ste13p[69]. Likewise, we constructed an artificial type II transmembrane Golgi protein GOTS_CyBBE which led to the improvement of (S)-SCO production by 33% ($p = 0.0001$). These results highlighted that the ER and Golgi might be utilized as promising organelles for metabolic engineering of heterologous pathway. Due to the microenvironmental differences between plants and yeast, it is important to consider different strategies when re-localizing specific plant-based enzymes to distinct organelles in yeast.

We have confirmed that McDBOX2 catalyzed the last step for BZDAs biosynthesis in yeast. Due to the substrate promiscuity of STOXs in vitro activity assays[25], we also explored McDBOX2 function on other substrates, by searching potential products in strains XJ07438 (XJ0743 + McDBOX2) with LC-MS/MS analysis. We observed two peaks, corresponding to m/z 324.13 [M]$^+$, and m/z 322.11 [M]$^+$, respectively, which might be the products derived from (S)-cheilanthifoline and (S)-stylopine via two-electron oxidation to form one N7 = C8 band

(Supplementary Fig. 30a–e). Combined with the results of 13,14-dihydropalmatine, 13,14-dihydrocolumbamine, 13,14-dihydroscoulerine and 13,14-dihydroberberine production catalyzed by McDBOX2 (Supplementary Fig. 21b, e, g and d), we assumed that McDBOX2 has a very broad substrate scope of tetrahydroprotoberberine alkaloids to produce 13,14-dihydroprotoberberine alkaloids. As it is known, the fragmentation way of protoberberines was not the retro-Diels-Alder cleavage, due to their unsaturated C-ring structure (Supplementary Fig. 26a)[70,71], which was mostly consistent with MS2 spectrum of dihydroprotoberberine alkaloids produced in this study (Supplementary Figs. 21c, f, h, 23e and 26b). Although we have identified that McDBOX2 could catalyze tetrahydroprotoberberine to produce corresponding 13,14-dihydroprotoberberine by forming one double band in yeast, we didn't manage to express functional STOX catalyzing tetrahydroprotoberberine to protoberberine by forming two double bands directly. Our results indicated that a possibility of catalyzation from tetrahydroprotoberberine to the corresponding protoberberine was consecutively accomplished by two enzymes (Supplementary Fig. 27a and b). This may provide new insights for screening additional STOXs to complete the biotransformation, thus realizing biological production of quaternary protoberberine.

In addition, the intermediates profiling from strain XJ0743 showed that three additional peaks, corresponding to m/z 342.18 [M]$^+$, 340.16 [M]$^+$ and m/z 338.14 [M]$^+$, respectively, were assumed to be the products of (S)-SCO, (S)-cheilanthifoline and (S)-Stylopine N-methylation by PsTNMT catalyzation (Supplementary Fig. 33b–g). Likewise, two peaks corresponding to m/z 354.18 [M]$^+$ and m/z 356.19 [M]$^+$ appeared in strain XJ0741, likely derived from (S)-canadine and (S)-THC undergoing N-methylation by PsTNMT catalyzation (Supplementary Fig. 33h–k). These findings provided experimental evidence that PsTNMT could accept (S)-SCO, (S)-cheilanthifoline, (S)-Stylopine, (S)-canadine and (S)-THC as substrates for N-methylation (Supplementary Fig. 33a), confirming a broad substrates scope of PsTNMT[18,28]. Our engineered yeast strains could biosynthesize a series of BIA compounds, some of which might not be found in plants due to localization-specific enzymes or potential transporters involved in the biosynthetic processes.

In contrast to several reports on the difficulties to realize the final step of oxidation in protoberberine and BZDAs biosynthesis in yeast[18,19,58,72], one recent study reported the expression of one oxidase FADOX5 from *P. somniferum* for sanguinarine production in *S. cerevisiae*[73]. However, the same enzyme exhibited smear expression in yeast in our study and resulted in comparable sanguinarine titer in strain XJ07434 relative with the control strain XJ07437 lacking oxidase candidate (Supplementary Fig. 28c and e). Production of sanguinarine without the expression of DBOX enzyme has been observed before, due to spontaneous and/or promiscuous enzyme activities in yeast[18,19], however, the production level was significantly lower than that McDBOX2 catalyzation (Supplementary Fig. 28e).

In this study, our (S)-RET platform strain can be used for scale-up production of many other natural BIAs, including compounds of the morphine family. Compared to vacuolar expression of CyBBE, ER lumenally expressed CyBBE_ERTS increased the intermediate (S)-SCO titer by more than 200% ($p < 0.0001$) in strain XJ0695, demonstrating its potential as a platform strain for highly producing the anticancer noscapine. The introduction of McDBOX2 completed the reconstruction of BZDAs pathway, allowing to biosynthesize chelerythrine, sanguinarine and chelirubine, with chelerythrine reaching a titer of 38.1 mg/L in yeast. Due to wide substrates specificity of McDBOX2 and PsTNMT, we observed the production of several BIAs in our engineered yeast, which can be used to explore their structure and medicinal activities. Furthermore, we envision that our final strains producing chelerythrine and chelirubine will facilitate the screening of downstream pathway enzymes to biosynthesize other pharmacological BZDAs, such as chelilutine and macarpine.

## Methods

### Strains and reagents

All strains used in this work are listed in Supplementary Data 1. *Escherichia coli* DH5α strain was used for routine plasmid assembly. All yeast strains were derived from IMX581 (*MATa ura3-52 can1Δ:: cas9-natNT2 TRP1 LEU2 HIS3, CEN.PK113-5D*).

YPD medium, consisting of $20\,g\,L^{-1}$ peptone (Difco), $10\,g\,L^{-1}$ yeast extract (Merck Millipore), and $20\,g\,L^{-1}$ glucose (VWR), was used for routine yeast culture and competent cells preparation. Synthetic dropout minus uracil medium (SD-Ura) ($6.7\,g\,L^{-1}$ yeast nitrogen base (YNB) without amino acids (Formedium), $0.77\,g\,L^{-1}$ complete supplement mixture without uracil (CSM-URA, Formedium) and $20\,g\,L^{-1}$ glucose (VWR)) was used for screening positive colonies transformed with *URA3* marker plasmid. CSM + 5-FOA medium, containing $6.7\,g\,L^{-1}$ YNB without amino acids, $0.79\,g\,L^{-1}$ complete supplement mixture (CSM, Formedium) and $0.8\,g\,L^{-1}$ 5-fluoroorotic acid (5-FOA, Sigma) was used for recycling the *URA3* marker. When necessary for preparing plates, $20\,g\,L^{-1}$ agar (Merck Millipore) was added. Delft media, containing $7.5\,g\,L^{-1}$ $(NH_4)_2SO_4$, $14.4\,g\,L^{-1}$ $KH_2PO_4$, $0.5\,g\,L^{-1}$ $MgSO_4 \cdot 7H_2O$, $20\,g\,L^{-1}$ glucose, $2\,ml\,L^{-1}$ trace metal solutions and $1\,ml\,L^{-1}$ vitamin solutions[74], supplemented with $60\,mg\,L^{-1}$ uracil if required, was used for shake flask fermentation for producing BIAs.

Gibson Assembly Kit and Golden Gate Assembly Kit were supplied from NEB. PrimeSTAR HS polymerase and Phusion polymerase were bought from Takara and ThermoFisher, respectively. Authentic standards dopamine hydrochloride (Sigma), (S)-reticuline (Cymit Pamplona, Spain), 4-hydroxyphenylacetic acid (4-HP acid) (TCI), 2-(4-hydroxyphenyl) ethanol (tyrosol) (TCI) were supplied. (S)-(-)-norco-claurine hydrobromide, (S)-scoulerine, (-)-tetrahydrocolumbamine, chelerythrine chloride and sanguinarium chloride were purchased from Toronto Research Chemicals (Toronto, Canada). (S)-canadine and berberine were supplied from Sigma. 7,8-dihydropalmatine and 7,8-dihydroberberine were bought from Yuanye (Shanghai China). Some intermediates of pathway were gifted from Guo Juan' lab (National Resource Center for Chinese Materia Medica, Beijing, China).

### Genetic manipulations

All primers used in this work are listed in Supplementary Data 2. IMX581 carrying chromosomally integrated Cas9 gene under TEF1 promoter, was employed as starting strain for all genetic manipulations. CRISPR-Cas9 based genome editing method was used for chromosomal gene deletion, insertion, and promoter replacement[75]. Promoters and terminators were amplified from IMX581 genome DNA by using PrimeSTAR HS or Phusion polymerase. The heterologous genes listed in Supplementary Data 3 were codon-optimized synthesized by GenScript. All primers were synthesized from Eurofins or Integrated DNA Technologies. For gene deletion, one 120 base pair (bp)-length oligo was designed, with 60 bp homologous to promoter and terminator of target gene, respectively. For gene insertion, upstream homologous arm, promoter, gene, terminator, and downstream homologous arm were first amplified with primers carrying 5′-terminal 30-40 bp overhangs. The entire repair fragment was assembled via overlapping PCR[76]. In cases where two or more genes were inserted into the one locus and the whole fragment was too long to assemble, the repair fragment was divided into several parts for construction. For promoter replacement, the candidate promoters with two-terminal 50 bp overhangs, identical to the upstream region of target promoter and target gene, respectively, were amplified from genome DNA. Specifically for the cassette design of ER-targeted CyBBE, we fused C-terminal HDEL peptides to CyBBE with a GSGS linker. In detail, we first amplified up-stream homologous arm XII-4 us, promoter CCW12p, fused gene sequence CyBBE_ERTS, terminator PGI1t, and down-stream homologous arm XII-4 ds by primer sets XII-4 us-F 571/ XII-4 us-CCW12p-R 572, CCW12p1-F/ CCW12p-DODC-R1, CCW12p-CyBBE-F 573/ BBE1-ERTS-PGI1t-R 920, BBE1-ERTS-PGI1t-F

921/ XII-5-PGI1t-r-F, and PGI1t-XII-4 ds-F 575/ XII-4 ds-R 576 (Supplementary Data 2) with yeast genome DNA or synthetized CyBBE sequence as templates. The whole cassette was assembled by overlap PCR and then transformed into the corresponding yeast strains along with pMEL10-XII-4-g RNA plasmid. All plasmids used in this work are listed in Supplementary Data 4.

To ease the assessment experiments in some cases, P416 plasmids were constructed for gene expression. Briefly, the vector scaffold was PCR amplified with p416_GPD plasmid as template, the candidate genes and vector scaffold were assembled into complete plasmids based on standard Gibson Assembly method.

For gRNA plasmid construction, either Gibson Assembly or Golden Gate Assembly was used. gRNAs were first predicted by using open-access tool (http://crispr.dbcls.jp/) for genes of target. pMEL10 was employed as background plasmid carrying SNR52 promoter and gRNA scaffold. For the plasmid bearing one gRNA, we first linearized the starting vector by PCR amplification. Two reverse complementary 60 bp oligos, each one consisting of 20 bp homologous segment with SNR52 promoter, 20 bp gRNA sequence and 20 bp homologous fragment with gRNA scaffold, were synthesized from Eurofins. Aliquot of oligos were annealed for 10 min at 95 °C, then slowly cold down to room temperature. The annealed oligo mix and linearized vector were assembled to complete plasmid. For the plasmid harboring two or more gRNAs, we first constructed host plasmid pMEL10-BsaI carrying only two BsaI cutting sites between SNR52 promoter and gRNA scaffold by Gibson Assemble. Briefly, we divided linearized vector pMEL10 into two parts by PCR amplification with the aim of removal of one BsaI site located in the antibiotic marker ampicillin sequence. An unrelated sequence to yeast genome, with one BsaI site adding to each terminus, was amplified. The three fragments were assembled into host plasmid pMEL10-BsaI. The commonly used template gRNA-SNR52p was obtained by overlap PCR. Based on this common template, we then obtained insertion fragments BsaI-N20(1)-gRNA-SNR52p-N20(2)-BsaI, … BsaI-N20(n-1)-gRNA-SNR52p-N20(n)-BsaI by different primers amplification. These insertion fragments and host plasmid pMEL10-BsaI were assembled into target plasmid with a certain number of gRNAs by Golden Gate Assembly. After sequencing all plasmids by Eurofins, the positive one was selected for further work. High-efficiency yeast transformation was performed by using the LiAc/SS carrier DNA/PEG method[77].

### Strain cultivation and metabolite quantification

Three or more biologically independent colonies were inoculated into $1\,mL$ delft media supplemented with $60\,mg\,L^{-1}$ uracil within $14\,mL$ culture tubes for 18-24 h at 30 °C, 220 rpm. The preculture was then transferred into $20\,mL$ fresh delft media supplemented with $60\,mg\,L^{-1}$ uracil within $125\,mL$ shake flakes with the initial $OD_{600}$ 0.05. Unless otherwise specified, the culture was kept for 72 h. For strains carrying P416 plasmid, the cultivation process was identical to strains without plasmid, except that the delft media without $60\,mg\,L^{-1}$ uracil was used.

After fermentation, $OD_{600}$ was measured, and metabolites were extracted from culture. Briefly, $500\,\mu L$ of culture was mixed with $500\,\mu L$ of 30% acetonitrile (ACN), vortexed thoroughly, and then centrifuged for 5 min at $13000\,g$. If necessary, the supernatant was further diluted with 15% ACN to the suitable concentration for LC-MS/MS analysis. To perform the conversion of tetrahydroprotoberberine to the corresponding quaternary protoberberine though heating, the culture was transferred to PCR tubes, and heated at 98 °C in PCR instrument. The samples were then stored at -20 °C until analysis.

One microliter of each sample was injected for analysis by LC-MS/MS system, consisting of a Shimadzu Nexera UHPLC system and a high-end hybrid triple quadrupole ion trap instrument (Sciex QTRAP 6500 + ) with Luna Omega 1.6 μm Polar C18 100 Å column (Phenomenex). Analytes were eluted at a constant flow rate of $400\,\mu L\,min^{-1}$ with 0.1% formic acid as solvent A and ACN with 0.1% formic acid as solvent

B. For analytes tyrosol, 4-HPAC, dopamine, (S)-NOR and (S)-RET, the following MS parameters were used: Curtain Gas (50); Collision Gas (Medium); IonSpray Voltage: -4500 V (4-HP acid and Tyrosol), 4500 V (Dopamine, (S)-NOR and (S)-RET); Temperature (450 °C). The gradient method was used as follows: 2% B to 10% B from 0-4 min, 10% B to 85% B from 4-6 min, held at 85% B from 6-7 min, 85% B to 2% B from 7−7.1 min, and held at 2% B from 7.1−9 min. (S)-SCO, intermediates of module VI, VII and VIII, chelerythrine and sanguinarine were separated with the following gradient method: 10% B from 0-0.1 min, 10% B to 45% B from 0.1-5 min, 45% B to 90% B from 5−5.5 min, held at 90% B from 5.5−7 min, 90% B to 10% B from 7−7.01 min, and held at 10% B from 7.01-9.5 min by using the identical column, flow rate and mobile phases. The source parameters were set as follows: Curtain Gas (30); Collision Gas (High); IonSpray Voltage (4500 V); Temperature (600 °C); Ion Source Gas 1 and 2 (60). Compounds from module IV and V were separated with the following gradient method: 10% B from 0−0.1 min, 10% B to 25% B from 0.1−6.0 min, 25% B to 70% B from 6.0−7.0 min, 70% B to 98% B from 7.0−7.1 min, held at 98% B from 7.1-8.5 min, 98% B to 10% B from 8.5−8.6 min, and held at 10% B from 8.6−10 min by using the identical column, flow rate and mobile phases. The source parameters were set as follows: Curtain Gas (40); Collision Gas (Medium); IonSpray Voltage (3000 V); Temperature (500 °C); Ion Source Gas 1 (40) and 2 (50). For quantification of analytes, MRM mode was used to monitor the transitions based on authentic standards. For qualification of analytes, Q3 MI mode or MS2 mode were used (Supplementary Table 2). Analyst and MultiQuant 3.0.3 software were used for data acquisition and processing.

Unknown compounds analysis was performed by using a Waters Xevo G2-X QTOF system with an electrospray ionization source operating in positive mode. Full scanning was conducted over an $m/z$ range of 100-800. The scanning time was 0.5 s, the detection time was 30 min, the low energy impact voltage was 6 V, and the high energy impact voltage was 15-40 V. Nitrogen gas was used as the solvent gas, and leucine enkephalin was used for real-time correction. All the target compounds were eluted at a constant flow rate of 300 μL min$^{-1}$ by using column HSS T3 Column, 100 Å, 1.8 μm, 2.1 mm×100 mm, with 0.04% formic acid as solvent A and methanol with 0.04% formic acid as solvent B. The gradient method was used as followed: 5% B to 10% B from 0-5.0 min, 10% B to 15% B from 5.0-20.0 min, 15% B to 25% B from 20.0-23.0 min, 25% B to 40% B from 23.0-26.0 min, 40% B to 90% B from 26.0-27.0 min, held at 90% B from 27.0-28.5 min, 90% B to 5% B from 28.5-28.6 min, and held at 5% B from 28.6-30 min. MassLnx software was used for data acquisition and processing.

## Western blot

The strains were cultured in 1 mL delft media for overnight at 30 °C, 220 rpm. The preculture was then transferred with 1:50 dilution into 5 mL fresh delft media and grown until OD$_{600}$ around 1. Then 3 OD cells were pelleted, washed twice with Phosphate buffered saline (PBS). 200 μL of acid-washed glass beads and 300 μL of PBS with 1x protease inhibitor cocktail (Sigma) were added to resuspend the pellet, and then cells were broken by running FastPrep according to manufacturer instructions. For the removal of potential N-linked oligosaccharide, cell lysates were first processed in denaturing buffer (0.5% SDS and 40 mM DTT) for 10 min at 98 °C, and subsequently treated with amidase PNGase F (ThermoFisher) for 1 h at 37 °C.

Samples, added with 4 X NuPAGE™ LDS Sample Buffer (Thermo-Fisher), were boiled for 10 min at 95 °C. 10 μL of samples were loaded into 4−20% Mini-PROTEAN® TGX Stain-Free™ Protein Gels (Bio-Rad), and run for 1 h at 150 V. Proteins were transferred onto Trans-Blot® Turbo™ Mini PVDF Transfer Packs (Bio-Rad) and blocked 2 h at room temperature in PBST (PBS + 0.1% Tween 20) with 5% milk. The membrane was washed three times with PBST for 5 min and incubated for 1 h at room temperature with 6x-His Tag Monoclonal Antibody (HIS.H8) (ThermoFisher), or Anti-GAPDH Antibody (G-9) (Santa Cruz

Biotechnology). The membrane was washed three times with PBST for 5 min and incubated for 1 h at room temperature with HRP conjugated Anti-Mouse IgG (H + L) Secondary Antibodies (Invitrogen). The membrane was washed three times with PBST for 5 min and incubated for 5 min with West Pico plus HRP substrate (ThermoFisher), and then analyzed with a ChemiDoc XRS image analyzer (Bio-Rad).

## RT-qPCR

To test the transcription level of CyBBE and its two variants, we performed RT-qPCR. Briefly, around 1 OD$_{600}$ cells were harvested by centrifugation at 12,000 × g for the RNA extraction using the RNeasy® kit from QIAGEN following the manufacturer's protocol. cDNA was synthesized by using QuantiTect Reverse Transcription kit (QIAGEN). The PCR reactions were done using the DyNAmo Flash SYBR® Green qPCR kit (Thermo Fisher Scientific) in Mx3005P qPCR system (Agilent). Gene *ACT1* acted as a reference to normalize RNA levels.

## In vitro activity assays

To test the activities of CyBBE and CyBBE_ERTS in distinct pH conditions in vitro, we processed the yeast cells with the identical method for western blot. subsequently, the lysate was washed twice with 100 mM tris-HCl buffer (pH 7.5) at 20000 x g for 5 min under 4 °C, and then suspended in citrate-phosphate buffer[79] or tris-HCl buffer with various pH. Adding commercial standard (S)-reticuline to keep the final concentration 5 mg/L, and processing the mix for 2 h under 37 °C. The dilutions were analyzed by LC-MS/MS later.

## Fluorescence microscopy analysis

To investigate the localization of CyBBE and McDBOX2 variants, green fluorescent protein (GFP) was fused to the C-terminus of each variant, and three marker genes, encoding ER protein Elo3, vacuole protein Vph1 and Golgi protein Chs5, respectively, were fused with a C-terminal red fluorescent protein (mBuRy2). The corresponding plasmids (see Supplementary Data 1 for detailed information) were transformed into wild type IMX581. The resultant strains were cultured in delft media for 18 h at 30 °C, 220 rpm. 2 μL of cells were spotted onto plain glass slides and visualized with a high-resolution confocal microscopy (The Carl Zeiss LSM 980 with Airyscan 2). To examine the co-localization of multiple fluorescent proteins inside encapsulin compartments, two splitted venus fragments (VN and VC) and miRFP670 fused with a C-terminal targeting peptide were incorporated into strain expressing shell protein EncA. The resultant strains were cultured in delft media for 18 h at 30 °C, 220 rpm. Subsequently, these yeast cells were visualized with a LEICA DM2000 microscope (Leica Microsystems CMS GmbH).

## Molecular dynamic simulations

Alphafold2.2.3 was first used to construct the three-dimensional structures of CyBBE, CyBBEGSGSHDEL and CyBBEGSGSVIML, which were subsequently optimized by using UCSF Chimera. The atomic charges of the proteins were calculated under AMBER14SB force field, and protonation states were assigned with the H + +3 online tool. The structures of the small molecule (S)-reticuline and the cofactor FAD were obtained from the PubChem database. The conformational sampling was performed with the open-source cheminformatics software package RDKit. The conformations were optimized in the MMFF94 force field, and the AM1-BCC partial charges were assigned by using UCSF Chimera.

Molecular docking experiments were performed by using AutoDock4.2 software. The optimal binding sites, predicted with SiteMap software, were set as the docking center. The coordinates of the docking center were set as center x = 6.09, center y = −0.12, and center z = −3.63. The box size was set to a 22.5 Å cubic box, and the spacing step size was set to 0.375. The maximum limitation for conformational search was set to 10,000. The conformational sampling and scoring were performed by using Lamarckian Genetic Algorithm

(LGA). The optimal complex of substrate, cofactor and protein were selected as initial models for subsequent dynamic simulation studies.

The open-source software package GROMACS5.1.5 was used to perform simulations. The simulation system was set in a closed environment with the temperature at 30 °C and pH to 5.5. The pressure in all systems was set to 1 bar. The complexes were set at the center of the simulation system under periodic boundary conditions, and the minimum distance from the edge of the protein to the edge of the box was set to 0.1 nm. The pdb2gmx was used to convert the receptor structure topology file into a recognized one by GROMACS under AMBEff14SB force field. The AmberTools was used to convert small molecules into topology files recognized by GROMACS, and the GAFF force field was used to parse ligand atoms. TIP3P water molecules were added to simulate the water environment. The steepest descent method was used to minimize the system energy for all atoms. In circumstance of protein position constrained, equilibrium simulations were carried out for 1000 ps under constant values of particles, volume, and temperature (NVT) and particles, pressure, and temperature (NPT), respectively. After NVT and NPT equilibration, all systems were subjected to 50 ns of production dynamics simulations, with simulations conducted every 2 fs. Covalent bond lengths were constrained by using the linear constraint solver algorithm, and long-range electrostatic interactions were treated according to the Particle Mesh Ewald (PME) method.

### Statistical analysis

At least three biological replicates were performed for each quantification, and the numerical values were depicted as means ± standard deviations. Statistical differences between control and derivative strains were assessed via unpaired $t$-tests using GraphPad Prism 8. In all cases, $P$-values < 0.05 were considered significant. * $p < 0.05$, ** $p < 0.01$ and *** $p < 0.001$.

### Reporting summary

Further information on research design is available in the Nature Portfolio Reporting Summary linked to this article.

## Data availability

Data supporting the findings of this work are available within the paper and its Supplementary Information files. A reporting summary for this Article is available as a Supplementary Information file. Source data are provided with this paper.

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

## Acknowledgements
We thank Mikael Molin for kindly providing the mPRDX4 plasmid, Shan Jiang and Centre for Cellular Imaging from University of Gothenburg for the help with fluorescent microscopes, and the Chalmers Mass Spectrometry Infrastructure for the assistance on metabolite analysis. This work was financially supported by Vetenskapsrådet (2018-06003), Stiftelsen för internationalisering av högre utbildning och forskning, National Key R&D Program of China (2020YFA0908000) and National Natural Science Foundation of China (82011530137 & 31961133007).

## Author contributions
Y.C. and X.J. conceived the study. X.J. performed most of the experiments. X.J., Y.C. analyzed all the experimental data and drafted the manuscript. X.F., J.B. and X.L. performed part of plasmids and yeast strain constructions, Q.L. and O.S. helped with LC-MS/MS analysis, J.G, L.H. and J.N. assisted with data analysis. All authors revised and approved the final version of the manuscript.

## Funding

## Competing interests
Y.C. and X.J. are inventors of pending patent applications (PCT/071270 and PCT/071276) arising from work on strategies for improved alkaloids production. Other authors declare no competing interests.
