## [Peer Review File · Nature Communications]

De novo production of protoberberine and benzophenanthridine alkaloids through metabolic engineering of yeast

Corresponding Author: Dr Yun Chen

This manuscript has been previously reviewed at another journal. This document only contains reviewer comments, rebuttal and decision letters for versions considered at Nature Communications.

Version 0:

Reviewer comments:

Reviewer #1

(Remarks to the Author)

The authors engineered *Saccharomyces cerevisiae* to de novo synthesize various protoberberine alkaloids and benzophenanthridine alkaloids, through employing the ER/cytosolic compartmentalization strategies. This manuscript presents numerous metabolic engineering strategies for protoberberines and BZDAs de novo synthesis and overproduction, which would be of interest to researchers in this field. However, the following comments should be addressed before publication.

1. The work highlights their engineering strategies applied to plant proteins. The authors mentioned that their ER retrograding strategy provides a new way for optimization of specific enzymes requiring post-translational processing. If this manuscript focused on protein engineering strategies, could they give more examples to show that this strategy is general applicable? And more discussions on the situations where this strategy could be applied could be added. How this was performed should be detailed described in the Method part.

2. For the conversion of tetrahydroprotoberberine to the corresponding quaternary protoberberine, the authors heated the cultures to increase the conversion efficiency. However, heating at such high temperature may hinder its downstream engineering and also is not feasible in future industrial applications. Are there any other strategies could be employed?

Version 1:

Reviewer comments:

Reviewer #1

(Remarks to the Author)

The authors have addressed the questions.

Responses to reviewers' comments

Reviewer #1 (Remarks to the Author):

The authors engineered *Saccharomyces cerevisiae* to de novo synthesize various protoberberine alkaloids and benzophenanthridine alkaloids, through employing the ER/cytosolic compartmentalization strategies. This manuscript presents numerous metabolic engineering strategies for protoberberines and BZDAs de novo synthesis and overproduction, which would be of interest to researchers in this field. However, the following comments should be addressed before publication.

1. The work highlights their engineering strategies applied to plant proteins. The authors mentioned that their ER retrograding strategy provides a new way for optimization of specific enzymes requiring post-translational processing. If this manuscript focused on protein engineering strategies, could they give more examples to show that this strategy is general applicable? And more discussions on the situations where this strategy could be applied could be added. How this was performed should be detailed described in the Method part.

Response: Thank you for this valuable comment. Our ER retrograding strategy provides a new way for optimization of specific enzymes requiring post-translational processing. In our study, the N-terminal signal peptide of CyBBE was proved to be essential for directing the nascent polypeptide undergoing processing steps for folding, post-translational modifications (i.e., glycosylation and disulfide bond formation). Therefore, commonly employed N-terminal engineering such as N-terminus truncations or N-terminal fusions would be unlikely to overcome the challenges associated with CyBBE and other similar examples. This novel strategy allows for (1) the BBE enzyme to pass through the secretory pathway, enabling proper folding and incorporation of key glycosylations; and (2) retrograde trafficking to the ER, where it exhibits optimal activity in a compartment with proper pH conditions (in contrast to vacuole). To the best of our knowledge, no studies utilizing this ER retrograding strategy were reported to improve the activity of target proteins. Although we can't provide more examples of this strategy, we believe that this strategy would be generally applicable for heterologous expression of specific enzymes requiring post-translational processing. In addition, each compartment has their unique physicochemical environment, such as pH and oxidative status, which may provide favorable conditions for the function of distinct enzymes. In yeast, it is known that vacuolar has a more acidic environment while the pH in ER is close to neutral. We now have shown that pH is an important factor rendering ER-localized BBE outperforming the vacuolar targeted BBE (Supplementary note 2 and Supplementary Fig. 16a-f). To the best of our knowledge, how the environmental conditions impact the function and activity of targeting enzymes was not fully evidenced. Therefore, this ER retrograding strategy provides a more favorable microenvironment (i.e., pH) for maximizing the catalytic activity of BBE, which could be applicable for expression of enzymes requiring proper pH conditions. We have now added discussions on the situations where this strategy could be applied (Line 548-556).

Following the reviewer's suggestion, we have added the detailed description on how to implement this strategy in the Method part. "Specifically for the cassette design of ER-targeted CyBBE, we fused C-terminal HDEL peptides to CyBBE with a GSGS linker. In detail, we first amplified up-stream homologous arm XII-4 us, promoter CCW12p, fused gene sequence CyBBE_ERTS, terminator PGI1t, and down-stream homologous arm XII-4 ds by primer sets XII-4 us-F 571/ XII-4 us-CCW12p-R 572, CCW12p1-F/ CCW12p-DODC-R1, CCW12p-CyBBE-F 573/ BBE1-ERTS-PGI1t-R 920, BBE1-ERTS-PGI1t-F 921/ XII-5-PGI1t-r-F, and PGI1t-XII-4 ds-F 575/ XII-4 ds-R 576 (Supplementary Data 2) with yeast genome DNA or synthesized CyBBE sequence as templates. The whole cassette was assembled by

overlap PCR and then transformed into the corresponding yeast strains along with pMEL10-XII-4-g RNA plasmid.” (Line 659-666)

2. For the conversion of tetrahydroprotoberberine to the corresponding quaternary protoberberine, the authors heated the cultures to increase the conversion efficiency. However, heating at such high temperature may hinder its downstream engineering and also is not feasible in future industrial applications. Are there any other strategies could be employed?

Response: We thank the reviewer for this comment. We completely agree with the reviewer’s opinion that heating at such high temperatures will not be an optimal solution for future industrial applications. Although some oxidizing agents, such as hydrogen peroxide and potassium persulfate, are probably able to oxidize tetrahydroprotoberberine to the corresponding quaternary protoberberine¹, this alternative is also cost-effective and unsustainable. In this study, we have evaluated all potential tetrahydroprotoberberine oxidases (STOXs) previously reported to convert tetrahydroprotoberberines to the corresponding quaternary protoberberines. In consistent with previous reports²⁻⁵, none of these enzymes could realize the final step of oxidation in protoberberine biosynthesis in yeast. Although we didn’t manage to express functional STOX catalyzing tetrahydroprotoberberine to produce protoberberine by forming two double bands directly, enzyme DBOX identified in our study could catalyze tetrahydroprotoberberine to produce 13,14-dihydroprotoberberine by forming one double band in yeast (Supplementary Figs. 27a and 27b). Our results indicated a possibility that the catalyzation from tetrahydroprotoberberine to the corresponding protoberberine was consecutively accomplished by two enzymes (Supplementary Figs. 27a and 27b). This may provide new insights for screening additional STOXs to complete the biotransformation, thus realizing bioproduction of quaternary protoberberines.

Supplementary Figure 27. The proposed pathway from (S)-tetrahydropalmatine to palmatine **a**, and from (S)-tetrahydroberberine to berberine **b**.

References

1. Jamil, O.K., Cravens, A., Payne, J.T., Kim, C.Y. & Smolke, C.D. Biosynthesis of tetrahydropapaverine and semisynthesis of papaverine in yeast. *Proc Natl Acad Sci U S A* **119**, e2205848119 (2022).
2. Fossati, E. et al. Reconstitution of a 10-gene pathway for synthesis of the plant alkaloid dihydrosanguinarine in *Saccharomyces cerevisiae*. *Nat Commun* **5**, 3283 (2014).
3. Trenchard, I.J. & Smolke, C.D. Engineering strategies for the fermentative production of plant alkaloids in yeast. *Metab Eng* **30**, 96-104 (2015).
4. Galanie, S. & Smolke, C.D. Optimization of yeast-based production of medicinal protoberberine alkaloids. *Microb Cell Fact* **14**, 144 (2015).
5. Han, J. & Li, S. De novo biosynthesis of berberine and halogenated benzyloquinoline alkaloids in *Saccharomyces cerevisiae*. *Commun Chem* **6**, 27 (2023).

Reviewer #1 (Remarks to the Author):

The authors have addressed the questions.

Response: We thank the reviewers for the constructive comments, which helped us to improve our manuscript.